# Certified Machine Unlearning
# via Noisy Stochastic Gradient Descent

**Eli Chien**
Department of Electrical and Computer Engineering
Georgia Institute of Technology
Georgia, U.S.A.
`ichien6@gatech.edu`

**Haoyu Wang**
Department of Electrical and Computer Engineering
Georgia Institute of Technology
Georgia, U.S.A.
`haoyu.wang@gatech.edu`

**Ziang Chen**
Department of Mathematics
Massachusetts Institute of Technology
Massachusetts, U.S.A.
`ziang@mit.edu`

**Pan Li**
Department of Electrical and Computer Engineering
Georgia Institute of Technology
Georgia, U.S.A.
`panli@gatech.edu`

## Abstract

"The right to be forgotten" ensured by laws for user data privacy becomes increasingly important. Machine unlearning aims to efficiently remove the effect of certain data points on the trained model parameters so that it can be approximately the same as if one retrains the model from scratch. We propose to leverage projected noisy stochastic gradient descent for unlearning and establish its first approximate unlearning guarantee under the convexity assumption. Our approach exhibits several benefits, including provable complexity saving compared to retraining, and supporting sequential and batch unlearning. Both of these benefits are closely related to our new results on the infinite Wasserstein distance tracking of the adjacent (un)learning processes. Extensive experiments show that our approach achieves a similar utility under the same privacy constraint while using $2\%$ and $10\%$ of the gradient computations compared with the state-of-the-art gradient-based approximate unlearning methods for mini-batch and full-batch settings, respectively.

## 1 Introduction

Machine learning models usually learn from user data where data privacy has to be respected. Certain laws, such as European Union's General Data Protection Regulation (GDPR), are in place to ensure "the right to be forgotten", which requires corporations to erase all information pertaining to a user if they request to remove their data. It is insufficient to comply with such privacy regulation by only removing user data from the dataset, as machine learning models can memorize training data information and risk information leakage [1, 2]. A naive approach to adhere to this privacy regulation is to retrain the model from scratch after every data removal request. Apparently, this approach is prohibitively expensive in practice for frequent data removal requests and the goal of machine unlearning is to perform efficient model updates so that the resulting model is (approximately)

38th Conference on Neural Information Processing Systems (NeurIPS 2024).

the same as retraining statistically. Various machine unlearning strategies have been proposed, including exact [3–6] and approximate approaches [7–11]. The later approaches allow a slight misalignment between the unlearned model and the retraining one in distribution under a notion similar to Differential Privacy (DP) [12].

The most popular approach for privatizing machine learning models with DP guarantee is arguably noisy stochastic gradient methods including the celebrated DP-SGD [13]. Mini-batch training is one of its critical components, which not only benefits privacy through the effect of privacy amplification by subsampling [14] but also provides improved convergence of the underlying optimization process. Several recent works [9, 11] based on full-batch (noisy) gradient methods may achieve certified approximate unlearning. Unfortunately, their analysis is restricted to the full-batch setting and it is non-trivial to extend these works to the mini-batch setting with tight approximate unlearning guarantees. The main challenge is to incorporate the randomness in the mini-batch sampling into the sensitivity-based analysis [9] or the Langevin-dynamics-based [11] analysis.

We aim to study mini-batch noisy gradient methods for certified approximate unlearning. The high-level idea of our unlearning framework is illustrated in Figure 1. Given a training dataset $\mathcal{D}$ and a fixed mini-batch sequence $\mathcal{B}$, the model first learns and then unlearns given unlearning requests, both via the projected noisy stochastic gradient descent (PNSGD). For sufficient learning epochs, we prove that the law of the PNSGD learning process converges to a *unique* stationary distribution $\nu_{\mathcal{D}|\mathcal{B}}$ (Theorem 3.1). When an unlearning request arrives, we update $\mathcal{D}$ to an adjacent dataset $\mathcal{D}'$ so that the data point subject to such request is removed. The approximate unlearning problem can then be viewed as moving from the current distribution $\nu_{\mathcal{D}|\mathcal{B}}$ to the target distribution $\nu_{\mathcal{D}'|\mathcal{B}}$ until $\varepsilon$-close in Rényi divergence for the desired privacy loss $\varepsilon$[1].

Our key observation is that the results of Altschuler and Talwar [15, 16], which study the convergence of PNSGD under the (strong) convexity assumption, can be leveraged after we formulate the approximate unlearning as above. They show that the Rényi divergence of two PNSGD processes with the same dataset but different initial distributions decays at a geometric rate, starting from the infinite Wasserstein distance ($W_\infty$) of initial distributions (Figure 1, step 3). As a result, if the initial $W_\infty$ distance can be properly characterized, we achieve the corresponding approximate unlearning guarantee by further taking the randomness of the mini-batches $\mathcal{B}$ into account (Figure 1, step 4). Therefore, the key step to establish the PNSGD-based unlearning guarantee is to characterize the initial $W_\infty$ distance of the unlearning process tightly.

The projection set diameter $2R$ for this $W_\infty$ distance adopted in [15] for DP analysis, unfortunately, leads to a vacuous unlearning guarantee, which cannot illustrate the computational advantage over the retraining from scratch. To alleviate this issue, we perform a careful $W_\infty$ distance tracking analysis along the adjacent PNSGD learning processes (Lemma 3.3), which leads to a much better bound $W_\infty(\nu_{\mathcal{D}|\mathcal{B}}, \nu_{\mathcal{D}'|\mathcal{B}}) \leq Z_\mathcal{B} \approx O(\eta M/b)$ (Figure 1, step 2) for bounded gradient norm $M$, mini-batch size $b$ and step size $\eta$. This ultimately leads to our unlearning guarantee $\varepsilon = O(Z_\mathcal{B}^2 c^{2Kn/b})$ for $K$ unlearning epochs and some rate $c < 1$. The computational benefit compared to the retraining from scratch naturally emerges by comparing two $W_\infty$ distances, $O(R)$ for the retraining from scratch and $O(\eta M/b)$ for our unlearning framework.

Our approach also naturally extends to multiple unlearning requests, including sequential and batch unlearning settings (Theorem 3.11 and Corollary J.1), by extending the $W_\infty$-tracking analysis (Lemma 3.4). Here, we may use the triangle inequality of $W_\infty$ distance, which leads to a tigher privacy loss bound (growing linearly to the number of unlearning requests) than that in the Langevin-dynamics-based analysis [11] via weak triangle inequality of Rényi divergence (growing exponentially).

Our results highlight the insights into privacy-utility-complexity trade-off regarding the mini-batch size $b$ for approximate unlearning. A smaller batch size $b$ leads to a better privacy loss decaying rate $O(c^{2Kn/b})$. However, an extremely small $b$ may degrade the model utility and incur instability. It also leads to a worse bound $Z_\mathcal{B} \approx O(\eta M/b)$, which degrades the computational benefits compared to retraining. We demonstrate such trade-off of our PNSGD unlearning results via experiments against the state-of-the-art full-batch ($b = n$) gradient-based approximate unlearning solutions [9, 11]. Our analysis provides a significantly better privacy-utility-complexity trade-off even when we restrict ourselves to $b = n$, and further improves the results by adopting mini-batches. Under the same

---

[1]We refer privacy loss as two-sided Rényi divergence of two distributions, which we defined as Rényi difference in Definition 2.1.

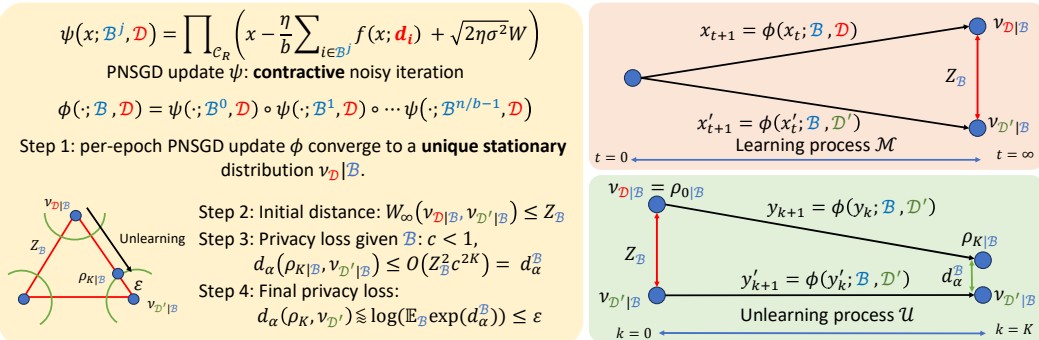

Figure 1: The overview of PNSGD unlearning. (Left) Proof sketch for PNSGD unlearning guarantees. (Right) PNSGD (un)learning processes on adjacent datasets. Given a mini-batch sequence $\mathcal{B}$, the learning process $\mathcal{M}$ induces a regular polyhedron where each vertex corresponds to a stationary distribution $\nu_{\mathcal{D}|\mathcal{B}}$ for each dataset $\mathcal{D}$. $\nu_{\mathcal{D}|\mathcal{B}}$ and $\nu_{\mathcal{D}|\mathcal{B}'}$ are adjacent if $\mathcal{D}, \mathcal{D}'$ differ in one data point. We provide an upper bound $Z_{\mathcal{B}}$ for the infinite Wasserstein distance $W_{\infty}(\nu_{\mathcal{D}|\mathcal{B}}, \nu_{\mathcal{D}|\mathcal{B}'})$, which is crucial for non-vacuous unlearning guarantees. Results of [16] allow us to convert the initial $W_{\infty}$ bound to Rényi difference bound $d_{\alpha}^{\mathcal{B}}$, and apply joint convexity of KL divergence to obtain the final privacy loss $\varepsilon$, which also take the randomness of $\mathcal{B}$ into account.

privacy constraint, our approach achieves similar utility while merely requiring $10\%, 2\%$ of gradient computations compared to baselines for full and mini-batch settings respectively.

## 1.1 Related Works

**Machine unlearning with privacy guarantees.** The concept of approximate unlearning uses a probabilistic definition of $(\epsilon, \delta)$-unlearning motivated by differential privacy [12], which is studied by [7, 8, 10]. Notably, the unlearning approach of these works involved Hessian inverse computation which can be computationally prohibitive in practice for high dimensional problems. Ullah et al. [5] focus on exact unlearning via a sophisticated version of noisy SGD. Their analysis is based on total variation stability which is not directly applicable to approximate unlearning settings and different from our analysis focusing on Rényi divergence. Neel et al. [9] leverage full-batch PGD for (un)learning and achieve approximate unlearning by publishing the final parameters with additive noise. Chien et al. [11] utilize full-batch PNGD for approximate unlearning with the analysis of Langevin dynamics. The adaptive unlearning requests setting is studied in [6, 17, 18], where the unlearning request may depend on the previous (un)learning results. It is possible to show that our framework is also capable of this adaptive setting since we do not keep any non-private internal states, though we only focus on non-adaptive settings in this work. We left a rigorous discussion for this as future work.

## 2 Preliminaries

We consider the empirical risk minimization (ERM) problem. Let $\mathcal{D} = \{\mathbf{d}_i\}_{i=1}^n$ be a training dataset with $n$ data point $\mathbf{d}_i$ taken from the universe $\mathcal{X}$. Let $f_{\mathcal{D}}(x) = \frac{1}{n}\sum_{i=1}^n f(x; \mathbf{d}_i)$ be the objective function that we aim to minimize with learnable parameter $x \in \mathcal{C}_R$, where $\mathcal{C}_R = \{x \in \mathbb{R}^d \mid \|x\| \leq R\}$ is a closed ball of radius $R$. We denote $\Pi_{\mathcal{C}_R} : \mathbb{R}^d \mapsto \mathcal{C}_R$ to be an orthogonal projection to $\mathcal{C}_R$. The norm $\|\cdot\|$ is standard Euclidean $\ell_2$ norm. $\mathcal{P}(\mathcal{C})$ is denoted as the set of all probability measures over a closed convex set $\mathcal{C}$. Standard definitions such as convexity are in Appendix D. We use $x \sim \nu$ to denote that a random variable $x$ follows the probability distribution $\nu$. We say two datasets $\mathcal{D}$ and $\mathcal{D}'$ are adjacent if they "differ" in only one data point. More specifically, we can obtain $\mathcal{D}'$ from $\mathcal{D}$ by *replacing* one data point. We next introduce a useful idea which we term as Rényi difference.

**Definition 2.1** (Rényi difference). Let $\alpha > 1$. For a pair of probability measures $\nu, \nu'$ with the same support, the $\alpha$ Rényi difference $d_{\alpha}(\nu, \nu')$ is defined as $d_{\alpha}(\nu, \nu') = \max\left(D_{\alpha}(\nu\|\nu'), D_{\alpha}(\nu'\|\nu)\right)$, where $D_{\alpha}(\nu\|\nu')$ is the $\alpha$ Rényi divergence defined as $D_{\alpha}(\nu\|\nu') = \frac{1}{\alpha-1}\log\left(\mathbb{E}_{x\sim\nu'}(\frac{\nu(x)}{\nu'(x)})^{\alpha}\right)$.

---

**Algorithm 1** (Un)learning with PNSGD

---

1: **Parameters:** stepsize $\eta$, noise standard deviation $\sigma$, dataset size $n$, mini-batch size $b$.
2: **Minibatch Generation:** randomly partition indices $[n]$ into $n/b$ mini-batches of size $b$: $\mathcal{B}^0, \ldots, \mathcal{B}^{n/b-1}$.
3: **Learning process** $\mathcal{M}(\mathcal{D})$**:** given a dataset $\mathcal{D} = \{\mathbf{d}_i\}_{i=1}^n \in \mathcal{X}^n$, sample initial parameter $x_0^0$ from a given initialization distribution $\nu_0$ supported on $\mathcal{C}_R$. The output is the last iterate $x_T^0$.
4: **for** epoch $t = 0, \ldots, T-1$ **do**
5:    **for** iteration $j = 0, \ldots, n/b - 1$ **do**
6:       $x_t^{j+1} = \Pi_{\mathcal{C}_R}\left(x_t^j - \eta g(x_t^j, \mathcal{B}^j) + \sqrt{2\eta\sigma^2}W_t^j\right)$, where $g(x_t^j, \mathcal{B}^j) = \frac{1}{b}\sum_{i \in \mathcal{B}^j}\nabla f(x_t^j; \mathbf{d}_i)$
       and $W_t^j \overset{\text{iid}}{\sim} \mathcal{N}(0, I_d)$.
7:    **end for**
8:    $x_{t+1}^0 = x_t^{n/b}$
9: **end for**
10: **Unlearning process** $\mathcal{U}(\mathcal{M}(\mathcal{D}), \mathcal{D}')$**:** given an updated dataset $\mathcal{D}' = \{\mathbf{d}_i'\}_{i=1}^n \in \mathcal{X}^n$ and current parameter $y_0^0$, the output is the last iterate $y_K^0$.
11: **for** epoch $k = 0, \ldots, K-1$ **do**
12:    **for** iteration $j = 0, \ldots, n/b - 1$ **do**
13:       $y_k^{j+1} = \Pi_{\mathcal{C}_R}\left(y_k^j - \eta g(y_k^j, \mathcal{B}^j) + \sqrt{2\eta\sigma^2}W_k^j\right)$, where $g(y_k^j, \mathcal{B}^j) = \frac{1}{b}\sum_{i \in \mathcal{B}^j}\nabla f(y_k^j; \mathbf{d}_i')$
       and $W_k^j \overset{\text{iid}}{\sim} \mathcal{N}(0, I_d)$.
14:    **end for**
15:    $y_{k+1}^0 = y_k^{n/b}$
16: **end for**

---

We are ready to introduce the formal definition of differential privacy and unlearning.

**Definition 2.2** (Rényi Differential Privacy (RDP) [19])**.** Let $\alpha > 1$. A randomized algorithm $\mathcal{M} : \mathcal{X}^n \mapsto \mathbb{R}^d$ satisfies $(\alpha, \varepsilon)$-RDP if for any adjacent dataset pair $\mathcal{D}, \mathcal{D}' \in \mathcal{X}^n$, the $\alpha$ Rényi difference $d_\alpha(\nu, \nu') \leq \varepsilon$, where $\mathcal{M}(\mathcal{D}) \sim \nu$ and $\mathcal{M}(\mathcal{D}') \sim \nu'$.

It is known to the literature that an $(\alpha, \varepsilon)$-RDP guarantee can be converted to the popular $(\epsilon, \delta)$-DP guarantee [12] relatively tight [19]. As a result, we will focus on establishing results with respect to $\alpha$ Rényi difference (and equivalently $\alpha$ Rényi difference). Next, we introduce our formal definition of unlearning based on $\alpha$ Rényi difference as well.

**Definition 2.3** (Rényi Unlearning (RU))**.** Consider a randomized learning algorithm $\mathcal{M} : \mathcal{X}^n \mapsto \mathbb{R}^d$ and a randomized unlearning algorithm $\mathcal{U} : \mathcal{R}^d \times \mathcal{X}^n \times \mathcal{X}^n \mapsto \mathcal{R}^d$. We say $(\mathcal{M}, \mathcal{U})$ achieves $(\alpha, \varepsilon)$-RU if for any $\alpha > 1$ and any adjacent datasets $\mathcal{D}, \mathcal{D}'$, the $\alpha$ Rényi difference $d_\alpha(\rho, \nu') \leq \varepsilon$, where $\mathcal{U}(\mathcal{M}(\mathcal{D}), \mathcal{D}') \sim \rho$ and $\mathcal{M}(\mathcal{D}') \sim \nu'$.

Our Definition 2.3 can be converted to the standard $(\epsilon, \delta)$-unlearning definition defined in [7–9], similar to RDP to DP conversion (see Appendix N). Since we work with the replacement definition of dataset adjacency, to unlearn a data point $\mathbf{d}_i$ we can simply replace it with any data point $\mathbf{d}_i' \in \mathcal{X}$ for the updated dataset $\mathcal{D}'$ in practice. One may also repeat the entire analysis with the objective function being the summation of individual loss for the standard add/remove notion of dataset adjacency. Finally, we will also leverage the infinite Wasserstein distance in our analysis.

**Definition 2.4** ($W_\infty$ distance)**.** The $\infty$-Wasserstein distance between distributions $\mu$ and $\nu$ on a Banach space $(\mathbb{R}^d, \|\cdot\|)$ is defined as $W_\infty(\mu, \nu) = \inf_{\gamma \in \Gamma(\mu,\nu)} \text{ess sup}_{(x,y)\sim\gamma} \|x - y\|$, where $(x, y) \sim \gamma$ means that the essential supremum is taken relative to measure $\gamma$ over $\mathbb{R}^d \times \mathbb{R}^d$ parametrized by $(x, y)$. $\Gamma(\mu, \nu)$ is the collection of couplings of $\mu$ and $\nu$.

## 2.1 Converting Initial $W_\infty$ Distance to Final Rényi Divergence Bound

An important component of our analysis is to leverage the result of [16], which is based on the celebrated privacy amplification by iteration analysis originally proposed in [20] and also utilized for DP guarantees of PNSGD in [15]. The goal of [16] is to analyze the mixing time of the PNSGD process, which can be viewed as the contractive noisy iterations due to the contractiveness of the gradient update under strong convexity assumption.

**Definition 2.5** (Contractive Noisy Iteration ($c$-CNI)). Given an initial distribution $\mu_0 \in \mathcal{P}(\mathbb{R}^d)$, a sequence of (random) $c$-contractive (equivalently, $c$-Lipschitz) functions $\psi_k : \mathbb{R}^d \mapsto \mathbb{R}^d$, and a sequence of noise distributions $\zeta_k$, we define the $c$-Contractive Noisy Iteration ($c$-CNI) by the update rule $X_{k+1} = \psi_{k+1}(X_k) + W_{k+1}$, where $W_{k+1} \sim \zeta_k$ independently and $X_0 \sim \mu_0$. We denote the law of the final iterate $X_K$ by $CNI_c(\mu_0, \{\psi_k\}, \{\zeta_k\})$.

**Lemma 2.6** (Metric-aware privacy amplification by iteration bound [20], simplified by [16] in Proposition 2.10). *Suppose $X_K \sim CNI_c(\mu_0, \{\psi_k\}, \{\zeta_k\})$ and $X'_K \sim CNI_c(\mu'_0, \{\psi_k\}, \{\zeta_k\})$ where the initial distribution satisfy $W_\infty(\mu_0, \mu'_0) \leq Z$, the update function $\psi_k$ are $c$-contractive, and the noise distributions $\zeta_k = \mathcal{N}(0, \sigma^2 I_d)$. Then we have*

$$D_\alpha(X_K \| X'_K) \leq \frac{\alpha Z^2}{2\sigma^2} \begin{cases} c^{2K} & \text{if } c < 1 \\ 1/K & \text{if } c = 1 \end{cases}. \tag{1}$$

Roughly speaking, Lemma 2.6 shows that if we have the $W_\infty$ distance of initial distributions of two PNSGD processes on the same dataset, we have the corresponding Rényi difference bound after $K$ iterations. The projection set diameter $2R$ is a default upper bound for $W_\infty$ distance if we do not care about the initial distributions as the case studied in [16] for mixing time analysis. In the unlearning scenario, the initial distributions of the unlearning processes, denoted by $\nu_{\mathcal{D}|\mathcal{B}}, \nu_{\mathcal{D}'|\mathcal{B}}$, are much more relevant. We can show a much tighter bound by analyzing $W_\infty(\nu_{\mathcal{D}|\mathcal{B}}, \nu_{\mathcal{D}'|\mathcal{B}})$ along the adjacent PNSGD learning processes.

## 3 Certified Unlearning Guaranatee for PNSGD

We start with introducing the (un)learning process with PNSGD with a cyclic mini-batch strategy (Algorithm 1). Note that this mini-batch strategy is not only commonly used for practical DP-SGD implementations in privacy libraries [21], but also in theoretical analysis for DP guarantees [22]. For the learning mechanism $\mathcal{M}$, we optimize the objective function with PNSGD on dataset $\mathcal{D}$ (line 3-8 in Algorithm 1). $\eta, \sigma^2 > 0$ are hyperparameters of step size and noise variance respectively. The initialization $\nu_0$ is an arbitrary distribution supported on $\mathcal{C}_R$ if not specified. For the unlearning mechanism $\mathcal{U}$, we fine-tune the current parameter with PNSGD on the updated dataset $\mathcal{D}'$ subject to the unlearning request (line 10-15 in Algorithm 1) with $y_0^0 = x_T^0 = \mathcal{M}(\mathcal{D})$. For the rest of the paper, we denote $\nu_t^j, \rho_k^j$ as the probability density of $x_t^j, y_k^j$ respectively. Furthermore, we denote $\mathcal{B} = \{\mathcal{B}^j\}_{j=0}^{n/b-1}$ the minibatch sequence described in Algorithm 1, where $b$ is the step size and we assume $n$ is divided by $b$ throughout the paper for simplicity[2]. We use $\nu_{\cdot|\mathcal{B}}$ to denote the conditional distribution of $\nu$. given $\mathcal{B}$.

### 3.1 Certified Unlearning Guarantees

Now we introduce the certified unlearning guarantees for PNSGD and the corresponding analysis illustrated in Figure 1. We first prove that for any fixed mini-batch sequence $\mathcal{B}$, the limiting distribution $\nu_{\mathcal{D}|\mathcal{B}}$ of the learning process exists, is unique, and stationery. The proof is deferred to Appendix E and is based on applying the results in [23] to establish the ergodicity of the learning process $x_t^0$.

**Theorem 3.1.** *Suppose that the closed convex set $\mathcal{C} \subset \mathbb{R}^d$ is bounded with $\mathcal{C}$ having a positive Lebesgue measure and that $\nabla f(\cdot; \mathbf{d}_i) : \mathcal{C} \to \mathbb{R}^d$ is continuous for all $i \in [n]$. The Markov chain $\{x_t := x_t^0\}$ in Algorithm 1 for any fixed mini-batch sequence $\mathcal{B}$ admits a unique invariant probability measure $\nu_{\mathcal{D}|\mathcal{B}}$ on the Borel $\sigma$-algebra of $\mathcal{C}$. Furthermore, for any $x \in \mathcal{C}$, the distribution of $x_t$ conditioned on $x_0 = x$ converges weakly to $\nu_{\mathcal{D}|\mathcal{B}}$ as $t \to \infty$, where $\nu_{\mathcal{D}|\mathcal{B}}$ is the conditional distribution of $\nu_{\mathcal{D}}$ given $\mathcal{B}$.*

Suppose the training epoch $T$ is large enough so that the model is well-trained for now, which means $\mathcal{M}(\mathcal{D}) \sim \nu_{\mathcal{D}|\mathcal{B}} = \rho_{0|\mathcal{B}}^0$ and our target "retraining distribution" is $\nu_{\mathcal{D}'|\mathcal{B}}$. Our goal is then to upper bound the Rényi difference $d_\alpha(\rho_{K|\mathcal{B}}^0, \nu_{\mathcal{D}'|\mathcal{B}})$ after $K$ unlearning epochs. In the case of insufficient training, the privacy loss is $d_\alpha(\rho_{K|\mathcal{B}}^0, \nu_{T|\mathcal{B}}^{0,\prime})$ where $\mathcal{M}(\mathcal{D}') \sim \nu_{T|\mathcal{B}}^{0,\prime}$ for $T$ training epochs. It can be upper bounded in terms of $d_\alpha(\rho_{K|\mathcal{B}}^0, \nu_{\mathcal{D}'|\mathcal{B}})$ and $d_\alpha(\nu_{T|\mathcal{B}}^{0,\prime}, \nu_{\mathcal{D}'|\mathcal{B}})$ via weak triangle inequality of

---

[2]When $n$ is not divided by $b$, we can simply drop the last $n - \lfloor n/b \rfloor b$ points.

Rényi divergence, which is provided in Theorem 3.2 below. We later provide a better bound by considering the randomness of $\mathcal{B}$.

**Theorem 3.2** (RU guarantee of PNSGD unlearning, fixed $\mathcal{B}$). *Assume $\forall \mathbf{d} \in \mathcal{X}$, $f(x; \mathbf{d})$ is L-smooth, M-Lipchitz and m-strongly convex in $x$. Let the learning and unlearning processes follow Algorithm 1 with $y_0^0 = x_T^0 = \mathcal{M}(\mathcal{D})$. Given any fixed mini-batch sequence $\mathcal{B}$, for any $\alpha > 1$, let $\eta \leq \frac{1}{L}$, the output of the $K^{th}$ unlearning iteration satisfies $(\alpha, \varepsilon)$-RU for any adjacent dataset $\mathcal{D}, \mathcal{D}'$, where*

$$\varepsilon \leq \frac{\alpha - 1/2}{\alpha - 1} \left( \varepsilon_1(2\alpha) + \varepsilon_2(2\alpha) \right), \varepsilon_1(\alpha) = \frac{\alpha(2R)^2}{2\eta\sigma^2} c^{2Tn/b}, \ \varepsilon_2(\alpha) = \frac{\alpha Z_{\mathcal{B}}^2}{2\eta\sigma^2} c^{2Kn/b},$$

$$Z_{\mathcal{B}} = W_\infty(\rho_{K|\mathcal{B}}^0, \nu_{\mathcal{D}'|\mathcal{B}}) \leq 2Rc^{Tn/b} + \min \left( \frac{1 - c^{Tn/b}}{1 - c^{n/b}} \frac{2\eta M}{b}, 2R \right),$$

*and $c = 1 - \eta m$.*

As we explained earlier, we need non-trivial $W_\infty$ bounds for adjacent PNSGD processes to obtain better unlearning guarantees when applying Lemma 2.6. We provide such results below and the proofs are deferred to Appendix L and M respectively.

**Lemma 3.3** ($W_\infty$ between adjacent PNSGD learning processes). *Consider the learning process in Algorithm 1 on adjacent datasets $\mathcal{D}$ and $\mathcal{D}'$ and a fixed mini-batch sequence $\mathcal{B}$. Assume $\forall \mathbf{d} \in \mathcal{X}$, $f(x; \mathbf{d})$ is L-smooth, M-Lipschitz and m-strongly convex in $x$. Let the index of different data point between $\mathcal{D}, \mathcal{D}'$ belongs to mini-batch $\mathcal{B}^{j_0}$. Then for $\eta \leq \frac{1}{L}$ and let $c = (1 - \eta m)$, we have*

$$W_\infty(\nu_{T|\mathcal{B}}^0, \nu_{T|\mathcal{B}}^{0,\prime}) \leq \min \left( \frac{1 - c^{Tn/b}}{1 - c^{n/b}} c^{n/b - j_0 - 1} \frac{2\eta M}{b}, 2R \right).$$

**Lemma 3.4** ($W_\infty$ between PNSGD learning process to its stationary distribution). *Following the same setting as in Theorem 3.2 and denote the initial distribution of the unlearning process as $\nu_0^0$. Then we have*

$$W_\infty(\nu_{T|\mathcal{B}}^0, \nu_{\mathcal{D}|\mathcal{B}}) \leq (1 - \eta m)^{Tn/b} W_\infty(\nu_0^0, \nu_{\mathcal{D}|\mathcal{B}}).$$

*Remark* 3.5. In [15], the authors used the default projection set diameter $2R$ as the $W_\infty$ distance upper bound for their DP results, see equations (3.5) and (5.2) therein. However, it yields a vacuous bound as an unlearning guarantee compared to retraining from scratch. Note that our tighter bound for $W_\infty$ distance is also useful for deriving later sequential unlearning guarantees compared to the prior work based on the analysis of Langevein dynamics [11]. Interestingly, this improved result can also be utilized for tightening the DP guarantee in [15] and make it more practically useful, as $2R$ can be very large in practice, which may be of independent interest.

We are ready to provide the sketch of proof for Theorem 3.2.

*Sketch of proof.* First note that the PNSGD update leads to a $(1 - \eta m)$-CNI process when $\eta \leq 1/L$ for any mini-batch sequence. Recall that $y_k^j$ is the unlearning process at epoch $k$ at iteration $j$, starting from $y_0^0 = x_T^0 = \mathcal{M}(\mathcal{D})$. Consider the "adjacent" process $y_k^{j,\prime}$ starting from $y_0^{0,\prime} = x_T^{0,\prime} = \mathcal{M}(\mathcal{D}')$ but still fine-tune on $\mathcal{D}'$ so that $y_k^j, y_k^{j,\prime}$ only differ in their initialization. Now, consider three distributions: $\nu_{\mathcal{D}|\mathcal{B}}, \nu_{T|\mathcal{B}}^0, \rho_{K|\mathcal{B}}^0$ are the stationary distribution for the learning processes, learning process at epoch $T$ and unlearning process at epoch $K$ respectively. Similarly, consider the "adjacent" processes that learn on $\mathcal{D}'$ and still unlearn on $\mathcal{D}'$ (see Figure 1 for the illustration). Denote distributions $\nu_{\mathcal{D}'|\mathcal{B}}, \nu_{T|\mathcal{B}}^{0,\prime}, \rho_{K|\mathcal{B}}^{0,\prime}$ for these processes similarly. Note that our goal is to bound $d_\alpha(\rho_{K|\mathcal{B}}^0, \nu_{T|\mathcal{B}}^{0,\prime})$ for the RU guarantee. By weak triangle inequality [19], we can upper bound it in terms of $d_{2\alpha}(\rho_{K|\mathcal{B}}^0, \nu_{\mathcal{D}'|\mathcal{B}})$ and $d_{2\alpha}(\nu_{\mathcal{D}'|\mathcal{B}}, \nu_{T|\mathcal{B}}^{0,\prime})$, which are the $\varepsilon_2$ and $\varepsilon_1$ terms in Theorem 3.2 respectively. For $d_\alpha(\nu_{\mathcal{D}'|\mathcal{B}}, \nu_{T|\mathcal{B}}^{0,\prime})$, we leverage the naive $2R$ bound for the $W_\infty$ distance between $\nu_{\mathcal{D}'|\mathcal{B}}, \nu_{0|\mathcal{B}}^{0,\prime}$ and applying Lemma 2.6 leads to the desired result. For $d_\alpha(\rho_{K|\mathcal{B}}^0, \nu_{\mathcal{D}'|\mathcal{B}})$, by triangle inequality of $W_\infty$ and Lemma 3.3, 3.4 one can show that the $W_\infty$ between $\rho_{0|\mathcal{B}}^0, \nu_{\mathcal{D}'|\mathcal{B}}$ is bounded by $Z_{\mathcal{B}}$ in Theorem 3.2. Further applying Lemma 2.6 again completes the proof.

*Remark* 3.6. Our proof only relies on the bounded gradient difference $\|\nabla f(x; \mathbf{d}) - \nabla f(x; \mathbf{d}')\| \leq 2M$ $\forall x \in \mathbb{R}^d$ and $\forall \mathbf{d}, \mathbf{d}' \in \mathcal{X}$ hence $M$-Lipschitz assumption can be replaced. In practice, we can leverage the gradient clipping along with a $\ell_2$ regularization for the convex objective function [22].

**The convergent case.** In practice, one often requires the model to be "well-trained", where a similar assumption is made in the prior unlearning literature [7, 8]. Under this assumption, we can further simplify Theorem 3.2 into the following corollary.

**Corollary 3.7.** *Under the same setting as of Theorem 3.2. When we additionally assume $T$ is sufficiently large so that $y_0^0 = M(\mathcal{D}) \sim \nu_{\mathcal{D}|\mathcal{B}}$. Then for any $\alpha > 1$ and $\eta \leq \frac{1}{L}$, the output of the $K^{th}$ unlearning iteration satisfies $(\alpha, \varepsilon)$-RU with $c = 1 - \eta m$, where*

$$\varepsilon \leq \frac{\alpha Z_\mathcal{B}^2}{2\eta\sigma^2} c^{2Kn/b}, \quad Z_\mathcal{B} = \min\left(\frac{1}{1 - c^{n/b}} \frac{2\eta M}{b}, 2R\right).$$

For simplicity, the rest of the discussion on our PNSGD unlearning will based on the well-trained assumption. From Corollary 3.7 one can observe that a smaller $b$ leads to a better decaying rate ($c^{2Kn/b}$) but also a potentially worse initial distance $Z_\mathcal{B} = O(1/((1 - c^{n/b})b))$. In general, choosing a smaller $b$ still leads to less epoch for achieving the desired privacy loss. In practice, choosing $b$ too small (e.g., $b = 1$) can not only degrade the utility but also incur instability (i.e., large variance) of the convergent distribution $\nu_{\mathcal{D}|\mathcal{B}}$, as $\nu_{\mathcal{D}|\mathcal{B}}$ depends on the design of mini-batches $\mathcal{B}$. One should choose a moderate $b$ to balance between privacy and utility, which is the unique privacy-utility-complexity trade-off with respect to $b$ revealed by our analysis.

**Computational benefit compared to retraining.** In the view of Corollary 3.7 or Lemma 2.6, it is not hard to see that a smaller initial $W_\infty$ distance leads to fewer PNSGD (un)learning epochs for being $\varepsilon$-close to a target distribution $\nu_{\mathcal{D}'|\mathcal{B}}$ in terms of Rényi difference $d_\alpha$. For PNSGD unlearning, we have provided a uniform upper bound $Z_\mathcal{B} = O(\eta M/((1 - c^{n/b})b))$ of such initial $W_\infty$ distance in Lemma 3.3. On the other hand, even if both $\nu_0, \nu_{\mathcal{D}}$ are both Gaussian with identical various and mean difference norm of $\Omega(1)$, we have $W_\infty(\nu_0, \nu_{\mathcal{D}'|\mathcal{B}}) = \Omega(1)$ for retraining from scratch. Our results show that a larger mini-batch size $b$ leads to more significant complexity savings compared to retraining. As we discussed above, one should choose a moderate size $b$ to balance between privacy and utility. In our experiment, we show that for commonly used mini-batch sizes (i.e., $b \geq 32$), our PNSGD unlearning is still much more efficient in complexity compared to retraining.

**Improved bound with randomized $\mathcal{B}$.** So far our results are based on a fixed (worst-case) mini-batch sequence $\mathcal{B}$. One can improve the privacy bound in Corollary 3.7 by taking the randomness of $\mathcal{B}$ into account under a non-adaptive unlearning setting. That is, the unlearning request is *independent* of the mini-batch sequence $\mathcal{B}$. See also our discussion in the related work. By taking the average of the bound in Corollary 3.7 in conjunction with an application of joint convexity of KL divergence [22], we can derive an improved guarantee beyond the worst-case of $\mathcal{B}$.

**Corollary 3.8.** *Under the same setting as of Theorem 3.2 but with random mini-batch sequences described in Algorithm 1. Then for any $\alpha > 1$, and $\eta \leq \frac{1}{L}$, the output of the $K^{th}$ unlearning iteration satisfies $(\alpha, \varepsilon)$-RU with $c = 1 - \eta m$, where $\varepsilon \leq \frac{1}{\alpha - 1} \log\left(\mathbb{E}_\mathcal{B} \exp\left(\frac{\alpha(\alpha - 1) Z_\mathcal{B}^2}{2\eta\sigma^2} c^{2Kn/b}\right)\right)$ and $Z_\mathcal{B}$ is the bound described in Lemma 3.3.*

**Different mini-batch sampling strategies.** We remark that our analysis can be extended to other mini-batch sampling strategies, such as sampling without replacement for each iteration. However, this strategy leads to a worse $Z_\mathcal{B}$ in our analysis of Lemma 3.3, which may seem counter-intuitive at first glance. This is due to the nature of the essential supremum taken in $W_\infty$. Although sampling without replacement leads to a smaller probability of sampling the index that gets modified due to the unlearning request, it is still non-zero for each iteration. Thus the worst-case difference $2\eta M/b$ between two adjacent learning processes in the mini-batch gradient update occurs at each iteration, which degrades the factor $1/(1 - c^{n/b})$ to $1/(1 - c)$ in Lemma 3.3. As a result, we choose to adopt the cyclic mini-batch strategy so that such a difference is guaranteed to occur only once per epoch and thus a better bound on $W_\infty$.

**Discussion on utility bound.** One can leverage the utility analysis in section 5 of [24] to derive the utility guarantee for the full batch setting $b = n$. We relegate the proof to Appendix K.

**Proposition 3.9.** *Under the same setting as Corollary 3.7 with $b = n$, $\eta \leq \frac{m}{2L^2}$ and assume the minimizer of any $f_\mathcal{D}$ is in the relative interior of $\mathcal{C}_R \subseteq \mathbb{R}^d$, for any given adjacent dataset pair $\mathcal{D}, \mathcal{D}'$ the output of the $K^{th}$ unlearning iteration $y_K^0$ satisfies*

$$\mathbb{E}[f_{\mathcal{D}'}(y_K^0)] - \inf_{x \in \mathcal{C}_R} f_{\mathcal{D}'}(x) \leq Mc^K \min(\frac{1}{1 - c} \frac{2\eta M}{n}, 2R) + \frac{2Ld\sigma^2}{m}. \tag{2}$$

Note that a similar analysis applies to both the mini-batch (i.e., $b \leq n$) and non-convergent case (i.e., Theorem 3.2) but the result is more complicated. We leave the rigorous analysis as future work. An important remark is that the second term $\frac{2Ld\sigma^2}{m}$ is the excess risk of $\nu_{\mathcal{D}'}$, which is controlled by the noise scale $\sigma$. This presents the privacy-utility trade-off as demonstrated in the DP scenario in [24].

**Without strong convexity.** Since Lemma 2.6 also applies to the convex-only case (i.e., $m = 0$ so that $c = 1$), repeating the same analysis leads to the following extension.

**Corollary 3.10.** *Under the same setting as of Theorem 3.2 but without strong convexity (i.e., $m = 0$). When we additionally assume $T$ is sufficiently large so that $y_0^0 = M(\mathcal{D}) \sim \nu_{\mathcal{D}|\mathcal{B}}$. Then for any $\alpha > 1$, and $\eta \leq \frac{1}{L}$, the output of the $K^{th}$ unlearning iteration satisfies $(\alpha, \varepsilon)$-RU, where*

$$\varepsilon \leq \frac{\alpha Z_{\mathcal{B}}^2}{2\eta\sigma^2}\frac{b}{Kn}, Z_{\mathcal{B}} = \min\left(\frac{2\eta MT}{b}, 2R\right).$$

There are several remarks for Corollary 3.10. First, the privacy loss now only decays linearly instead of exponentially as opposed to the strongly convex case. Second, $Z_{\mathcal{B}}$ now can grow linearly in training epoch $T$. As a result, the computational benefit of our approach compared to retraining may vanish for large $T$ such that $\frac{2\eta MT}{b} \geq 2R$. Nevertheless, the computational benefit against retraining persists for moderate $T$ such that $\frac{2\eta MT}{b} < 2R$. This condition can be met if the model learns reasonably well with moderate $T$ and the projection diameter $2R$ is not set to be extremely small. For example, with $2R = 10, b = 128, \eta = 1$ and $M = 1$, any training epoch $T < 640$ will lead to $\frac{2\eta MT}{b} < 2R$. Still, we conjecture a better analysis is needed beyond strong convexity.

### 3.2 Unlearning Multiple Data Points

So far we have focused on one unlearning request and unlearning one point. In practice, multiple unlearning requests can arrive sequentially (sequential unlearning) and each unlearning request may require unlearning multiple points (batch unlearning). Below we demonstrate that our PNSGD unlearning naturally supports sequential and batch unlearning as well.

**Sequential unlearning.** As long as we can characterize the initial $W_\infty$ distance for any mini-batch sequences, we have the corresponding $(\alpha, \varepsilon)$-RU guarantee due to Corollary 3.7. Thanks to our geometric view of the unlearning problem (Figure 2) and $W_\infty$ is indeed a metric, applying triangle inequality naturally leads to an upper bound on the initial $W_\infty$ distance. By combining Lemma 3.3 and Lemma 3.4, we have the following sequential unlearning guarantee.

**Theorem 3.11** ($W_\infty$ bound for sequential unlearning). *Under the same assumptions as in Corollary 3.7. Assume the unlearning requests arrive sequentially such that our dataset changes from $\mathcal{D} = \mathcal{D}_0 \to \mathcal{D}_1 \to \ldots \to \mathcal{D}_S$, where $\mathcal{D}_s, \mathcal{D}_{s+1}$ are adjacent. Let $y_k^{j,(s)}$ be the unlearned parameters for the $s^{th}$ unlearning request at $k^{th}$ unlearning epoch and $j^{th}$ iteration following Algorithm (1) on $\mathcal{D}_s$ and $y_0^{0,(s+1)} = y_{K_s}^{0,(s)} \sim \bar{\nu}_{\mathcal{D}_s|\mathcal{B}}$, where $y_0^{0,(0)} = x_\infty$ and $K_s$ is the unlearning steps for the $s^{th}$ unlearning request. For any $s \in [S]$, we have $W_\infty(\bar{\nu}_{\mathcal{D}_{s-1}|\mathcal{B}}, \nu_{\mathcal{D}_s|\mathcal{B}}) \leq Z_{\mathcal{B}}^{(s)}$, where $Z_{\mathcal{B}}^{(s+1)} = \min(c^{K_s n/b} Z_{\mathcal{B}}^{(s)} + Z_{\mathcal{B}}, 2R), \; Z_{\mathcal{B}}^{(1)} = Z_{\mathcal{B}}, Z_{\mathcal{B}}$ and $c$ are described in Corollary 3.7.*

By combining Corollary 3.7 and Theorem 3.11, we can establish the least unlearning iterations of each unlearning request $\{K_s\}_{s=1}^S$ to achieve $(\alpha, \varepsilon)$-RU simultaneously. Notably, our sequential unlearning bound is much better than the one in [11], especially when the number of unlearning requests is large. The key difference is that [11] have to leverage weak triangle inequality for Rényi divergence, which *double* the Rényi divergence order $\alpha$ for each sequential unlearning request. In contrast, since our analysis only requires tracking the initial

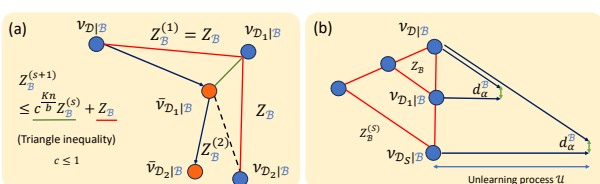

Figure 2: Illustration of (a) sequential unlearning and (b) batch unlearning. The key idea is to establish an upper bound on the initial $W_\infty$ distance. (a) For sequential unlearning, the initial $W_\infty$ distance bound $Z_{\mathcal{B}}^{(s)}$ for each $s^{th}$ unlearning request can be derived with triangle inequality. (b) For batch unlearning, we analyze the case that two learning processes can differ in $S \geq 1$ points.

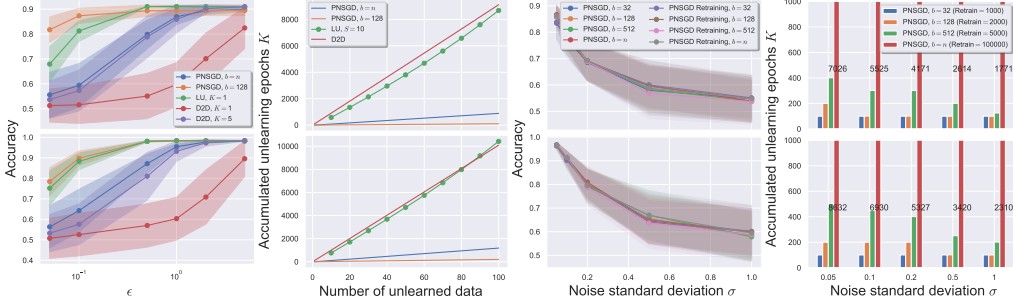

(a) Unlearning one point    (b) Unlearning 100 points    (c) Noise v.s. accuracy    (d) Noise v.s. complexity

Figure 3: Main experiments, where the top and bottom rows are for MNIST and CIFAR10 respectively. (a) Compare to baseline for unlearning one point using limited $K$ unlearning epoch. For PNSGD, we use only $K = 1$ unlearning epoch. For D2D, we allow it to use $K = 1, 5$ unlearning epochs. (b) Unlearning 100 points sequentially versus baseline. For LU, since their unlearning complexity only stays in a reasonable range when combined with batch unlearning of size $S$ sufficiently large, we report such a result only. (c,d) Noise-accuracy-complexity trade-off of PNSGD for unlearning 100 points sequentially with various mini-batch sizes $b$, where all methods achieve $(\epsilon, 1/n)$-unlearning guarantee with $\epsilon = 0.01$. We also report the required accumulated epochs for retraining for each $b$.

$W_\infty$ distance, where the standard triangle inequality can be applied. As a result, our analysis can better handle the sequential unlearning case. We also demonstrate in Section 4 that the benefit offered by our results is significant in practice.

**Batch unlearning.** We can extend Lemma 3.3 to the case that adjacent dataset $\mathcal{D}, \mathcal{D}'$ can differ in $S \geq 1$ points, which further leads to batch unlearning guarantee. We relegate the result in Appendix J.

## 4 Experiments

*Benchmark datasets.* We consider binary logistic regression with $\ell_2$ regularization. We conduct experiments on MNIST [25] and CIFAR10 [26], which contain 11,982 and 10,000 training instances respectively. We follow the setting of [7, 11] to distinguish digits 3 and 8 for MNIST so that the problem is a binary classification. For the CIFAR10 dataset, we distinguish labels 3 (cat) and 8 (ship) and leverage the last layer of the public ResNet18 [27] embedding as the data features, which follows the setting of [7] with public feature extractor.

*Baseline methods.* Our baseline methods include Delete-to-Descent (D2D) [9] and Langevin Unlearning (LU) [11], which are the state-of-the-art full-batch gradient-based approximate unlearning methods. Note that when our PNSGD unlearning chooses $b = n$ (i.e., full batch), the learning and unlearning iterations become PNGD which is identical to LU. Nevertheless, the corresponding privacy bound is still different as we leverage the analysis different from those based on Langevin dynamics in [11]. Hence, we still treat these two methods differently in our experiment. For D2D, we leverage Theorem 9 and 28 in [9] for privacy accounting depending on whether we allow D2D to have an internal non-private state. Note that allowing an internal non-private state provides a weaker notion of privacy guarantee [9] and both PNSGD and LU by default do not require it. We include those theorems for D2D and a detailed explanation of its possible non-privacy internal state in Appendix O. For LU, we leverage their Theorem 3.2, and 3.3 for privacy accounting [11], which are included in Appendix P.

All experimental details can be found in Appendix N, including how to convert $(\alpha, \varepsilon)$-RU to the standard $(\epsilon, \delta)$-unlearning guarantee. Our code is publicly available[3]. We choose $\delta = 1/n$ for each dataset and require all tested unlearning approaches to achieve $(\epsilon, \delta)$-unlearning with different $\epsilon$. We report test accuracy for all experiments as the utility metric. We set the learning iteration $T = 10, 20, 50, 1000$ to ensure PNSGD converges for mini-batch size $b = 32, 128, 512, n$ respectively. All results are averaged over 100 independent trials with standard deviation reported as shades in figures.

---

**Unlearning one data point with $K = 1$ epoch.** We first consider the setting of unlearning one data point using only one unlearning epoch (Figure 3a). For both LU and PNSGD, we use only $K = 1$ unlearning epoch. Since D2D cannot achieve a privacy guarantee with only limited (i.e., less than 10) unlearning epoch without a non-private internal state, we allow D2D to have it and set $K = 1, 5$ in this experiment. Even in this case, PNSGD still outperforms D2D in both utility and unlearning complexity. Compared to LU, our mini-batch setting either outperforms or is on par with it. Interestingly, we find that LU gives a better privacy bound compared to full-batch PNSGD ($b = n$) and thus achieves better utility under the same privacy constraint, see Appendix P for the detailed comparisons. Nevertheless, due to the use of weak triangle inequality in LU analysis, we will see that our PNSGD can outperform LU significantly for multiple unlearning requests.

**Unlearning multiple data points.** Let us consider the case of multiple (100) unlearning requests (Figure 3b). We let all methods achieve the same $(1, 1/n)$-unlearning guarantee for a fair comparison. We do not allow D2D to have an internal non-private state anymore in this experiment for a fair comparison. Since the privacy bound of LU only gives reasonable unlearning complexity with a limited number of sequential unlearning updates [11], we allow it to unlearn $S = 10$ points at once. We observe that PNSGD requires roughly $10\%$ and $2\%$ of unlearning epochs compared to D2D and LU for $b = n$ and $b = 128$ respectively, where all methods exhibit similar utility (0.9 and 0.98 for MNIST and CIFAR10 respectively). It shows that PNSGD is much more efficient compared to D2D and LU. Notably, while both PNSGD with $b = n$ and LU (un)learn with PNGD iterations, the resulting privacy bound based on our PABI-based analysis is superior to the one pertaining to Langevin-dynamic-based analysis in [11]. See our discussion in Section 3.2 for the full details.

**Privacy-utility-complexity trade-off.** We now investigate the inherent utility-complexity trade-off regarding noise standard deviation $\sigma$ and mini-batch size $b$ for PNSGD under the same privacy constraint, where we require all methods to achieve $(0.01, 1/n)$-unlearning guarantee for 100 sequential unlearning requests (Figure 3c and 3d). We can see that smaller $\sigma$ leads to a better utility, yet more unlearning epochs are needed for PNSGD to achieve $\epsilon = 0.01$. On the other hand, smaller mini-batch size $b$ requires fewer unlearning epochs $K$ as shown in Figure 3d, since more unlearning iterations are performed per epoch. Nevertheless, we remark that choosing $b$ too small may lead to degradation of model utility or instability. Decreasing the mini-batch size $b$ from 32 to 1 reduces the average accuracy of training from scratch from 0.87 to 0.64 and 0.97 to 0.81 on MNIST and CIFAR10 respectively for $\sigma = 0.03$. In practice, one should choose a moderate mini-batch size $b$ to ensure both good model utility and unlearning complexity. Finally, we also note that PNSGD achieves a similar utility while much better complexity compared to retraining from scratch, where PNSGD requires at most $1, 5$ unlearning epochs per unlearning request for $b = 32, 512$ respectively.

## 5 Limitations and Conclusion

**Limitation.** Since our analysis is built on the works of [15, 16], we share the same limitation that the (strong) convexity assumption is required. It is an open problem on how to extend such analysis beyond convexity assumption as stated in [15, 16]. While we resolve this open problem for establishing DP properties of PNSGD in our recent work [28], it is still unclear whether the same success can be generalized to the unlearning problem. One interesting direction is to leverage the Langevin dynamic analysis [29] instead as in [11], which can deal with non-convex problems in theory yet we conjecture the resulting bounds can be loose, and more complicated.

**Conclusion.** We propose to leverage projected noisy stochastic gradient descent (PNSGD) for machine unlearning problem. We provide its unlearning guarantees as well as many other algorithmic benefits of PNSGD for unlearning under the convexity assumption. Our results are closely related to our new results on infinite Wasserstein distance tracking of the adjacent (un)learning processes, which is also leveraged in our concurrent work for studying DP-PageRank algorithms [30]. Extensive experiments show that our approach achieves a similar utility under the same privacy constraint while using $2\%$ and $10\%$ of the gradient computations compared with the state-of-the-art gradient-based approximate unlearning methods for mini-batch and full-batch settings, respectively.

## Acknowledgments and Disclosure of Funding

The authors thank Sinho Chewi, Wei-Ning Chen, and Ayush Sekhari for the helpful discussions. E. Chien, H. Wang and P. Li are supported by NSF awards OAC-2117997 and JPMC faculty award.

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

# A   Additional related works

**Differential privacy of noisy gradient methods.** DP-SGD [13] is arguably the most popular method for ensuring a DP guarantee for machine learning models. Since it leverages the DP composition theorem and thus the privacy loss will diverge for infinite training epochs. Recently, researchers have found that if we only release the last step of the trained model, then we can do much better than applying the composition theorem. A pioneer work [31] studied the DP properties of Langevin Monte Carlo methods. Yet, they do not propose to use noisy GD for general machine learning problems. A recent line of work [22, 32] shows that PNSGD training can not only provide DP guarantees, but also the privacy loss is at most a finite value even if we train with an infinite number of iterations. The main analysis therein is based on the analysis of Langevin Monte Carlo [29, 33]. In the meanwhile, [15] also provided the DP guarantees for PNSGD training but with analysis based on privacy amplification by iteration [20]. None of these works study how PNSGD can also be leveraged for machine unlearning.

# B   Limitations

Since our analysis is built on the works of [15, 16], we share the same limitation that the (strong) convexity assumption is required. It is an open problem on how to extend such analysis beyond convexity assumption as stated in [15, 16]. While we resolve this open problem for the DP properties of PNSGD in our recent work [28], it is still unclear whether the same success can be generalized to the unlearning problem. One interesting direction is to leverage the Langevin dynamic analysis [29] instead as in [11], which can deal with non-convex problems in theory yet we conjecture the resulting bounds can be loose, and more complicated.

# C   Broader Impact

Our work study the theoretical unlearning guarantees of projected stochastic noisy gradient descent algorithm for convex problems. We believe our work is a foundational research and does not have a direct path to any negative applications.

# D   Standard definitions

Let $f : \mathbb{R}^d \mapsto \mathbb{R}$ be a mapping. We define smoothness, Lipschitzness, and strong convexity as follows:

$$L\text{-smooth: } \forall x, y \in \mathbb{R}^d, \ \|\nabla f(x) - \nabla f(y)\| \leq L\|x - y\| \tag{3}$$

$$m\text{-strongly convex: } \forall x, y \in \mathbb{R}^d, \ \langle x - y, \nabla f(x) - \nabla f(y) \rangle \geq m\|x - y\|^2 \tag{4}$$

$$M\text{-Lipschitzs: } \forall x, y \in \mathbb{R}^d, \ \|f(x) - f(y)\| \leq M\|x - y\|. \tag{5}$$

Furthermore, we say $f$ is convex means it is 0-strongly convex.

# E   Existence of limiting distribution

*Theorem.* Suppose that the closed convex set $\mathcal{C} \subset \mathbb{R}^d$ is bounded with $\mathcal{C}$ having a positive Lebesgue measure and that $\nabla f(\cdot; \mathbf{d}_i) : \mathcal{C} \to \mathbb{R}^d$ is continuous for all $i \in [n]$. The Markov chain $\{x_t := x_t^0\}$ in Algorithm 1 for any fixed mini-batch sequence $\mathcal{B} = \{\mathcal{B}^j\}_{j=1}^{n/b-1}$ admits a unique invariant probability measure $\nu_{\mathcal{D}|\mathcal{B}}$ on the Borel $\sigma$-algebra of $\mathcal{C}$. Furthermore, for any $x \in \mathcal{C}$, the distribution of $x_t$ conditioned on $x_0 = x$ converges weakly to $\nu_{\mathcal{D}|\mathcal{B}}$ as $t \to \infty$.

The proof is almost identical to the proof of Theorem 3.1 in [11] and we include it for completeness. We start by proving that the process $\{x_t := x_t^0\}$ admits a unique invariant measure (Proposition E.1) and then show that the process converges to such measure which is in fact a probability measure (Theorem E.2). Combining these two results completes the proof of Theorem 3.1.

**Proposition E.1.** *Suppose that the closed convex set $\mathcal{C} \subset \mathbb{R}^d$ is bounded with $Leb(\mathcal{C}) > 0$ where Leb denotes the Lebesgue measure and that $\nabla f(\cdot; \mathbf{d}_i) : \mathcal{C} \to \mathbb{R}^d$ is continuous for all $i \in [n]$. Then the*

*Markov chain $\{x_t := x_t^0\}$ defined in Algorithm 1 for any fixed mini-batch sequence $\mathcal{B} = \{\mathcal{B}^j\}_{j=1}^{n/b-1}$ admits a unique invariant measure (up to constant multiples) on $\mathcal{B}(\mathcal{C})$ that is the Borel $\sigma$-algebra of $\mathcal{C}$.*

*Proof.* This proposition is a direct application of results from [23]. According to Proposition 10.4.2 in [23], it suffices to verify that $\{x_t\}$ is recurrent and strongly aperiodic.

1. *Recurrency.* Thanks to the Gaussian noise $W_t$, $\{x_t\}$ is Leb-irreducible, i.e., it holds for any $x \in \mathcal{C}$ and any $A \in \mathcal{B}(\mathcal{C})$ with $\text{Leb}(A) > 0$ that

$$L(x, A) := \mathbb{P}(\tau_A < +\infty \mid x_0 = x) > 0,$$

   where $\tau_A = \inf\{t \geq 0 : x_t \in A\}$ is the stopping time. Therefore, there exists a Borel probability measure $\psi$ such that that $\{x_t\}$ is $\psi$-irreducible and $\psi$ is maximal in the sense of Proposition 4.2.2 in [23]. Consider any $A \in \mathcal{B}(\mathcal{C})$ with $\psi(A) > 0$. Since $\{x_t\}$ is $\psi$-irreducible, one has $L(x, A) = \mathbb{P}(\tau_A < +\infty \mid x_0 = x) > 0$ for all $x \in \mathcal{C}$. This implies that there exists $T \geq 0$, $\delta > 0$, and $B \in \mathcal{B}(\mathcal{C})$ with $\text{Leb}(B) > 0$, such that $\mathbb{P}(x_T \in A \mid x_0 = x) \geq \delta$, $\forall\, x \in B$. Therefore, one can conclude for any $x \in \mathcal{C}$ that

$$U(x, A) := \sum_{t=0}^{\infty} \mathbb{P}(x_t \in A \mid x_0 = x)$$

$$\geq \sum_{t=1}^{\infty} \mathbb{P}(x_{t+T} \in A \mid x_t \in B, x_0 = x) \cdot \mathbb{P}(x_t \in B \mid x_0 = x)$$

$$\geq \sum_{t=1}^{\infty} \delta \cdot \inf_{y \in \mathcal{C}} \mathbb{P}(x_t \in B \mid x_{t-1} = y)$$

$$= +\infty,$$

   where we used the fact that $\inf_{y \in \mathcal{C}} \mathbb{P}(x_t \in B \mid x_{t-1} = y) = \inf_{y \in \mathcal{C}} \mathbb{P}(x_1 \in B \mid x_0 = y) > 0$ that is implies by $\text{Leb}(B) > 0$ and the boundedness of $\mathcal{C}$ and $\bigcup_{i \in \mathcal{B}^{n/b}} \nabla f(\mathcal{C}; \mathbf{d}_i)$. Let us remark that we actually have compact $\bigcup_{i \in \mathcal{B}^{n/b}} \nabla f(\mathcal{C}; \mathbf{d}_i)$ since $\mathcal{C}$ is compact and $\nabla f(\cdot; \mathbf{d}_i)$ is continuous. The arguments above verify that $\{x_t\}$ is recurrent (see Section 8.2.3 in [23] for definition).

2. *Strong aperiodicity.* Since $\mathcal{C}$ and $\bigcup_{i \in \mathcal{B}^{n/b}} \nabla f(\mathcal{C}; \mathbf{d}_i)$ are bounded and the density of $W_t$ has a uniform positive lower bound on any bounded domain, there exists a non-zero multiple of the Lebesgue measure, say $\nu_1$, satisfying that

$$\mathbb{P}(x_1 \in A \mid x_0 = x) \geq \nu_1(A), \quad \forall\, x \in \mathcal{C},\; A \in \mathcal{B}(\mathcal{C}).$$

   Then $\{x_t\}$ is strongly aperiodic by the equation above and $\nu_1(\mathcal{C}) > 0$ (see Section 5.4.3 in [23] for definition).

The proof is hence completed. $\qquad\square$

**Theorem E.2.** *Under the same assumptions as in Proposition E.1, the Markov chain $\{x_t\}$ admits a unique invariant probability measure $\nu_{\mathcal{D}|\mathcal{B}}$ on $\mathcal{B}(\mathcal{C})$. Furthermore, for any $x \in \mathcal{C}$, the distribution of $x_t = x_t^0$ generated by the learning process in Algorithm 1 conditioned on $x_0 = x$ converges weakly to $\nu_{\mathcal{D}|\mathcal{B}}$ as $t \to \infty$.*

*Proof.* It has been proved in Proposition E.1 that $\{x_t\}$ is strongly aperiodic and recurrent with an invariant measure. Consider any $A \in \mathcal{B}(\mathcal{C})$ with $\psi(A) > 0$ and use the same settings and notations as in the proof of Proposition E.1. There exists $T \geq 0$, $\delta > 0$, and $B \in \mathcal{B}(\mathcal{C})$ with $\text{Leb}(B) > 0$, such that $\mathbb{P}(x_T \in A \mid x_0 = x) \geq \delta$, $\forall\, x \in B$. This implies that for any $t \geq 0$ and any $x \in \mathcal{C}$,

$$\mathbb{P}(x_{t+T+1} \in A \mid x_t = x) = \mathbb{P}(x_{T+1} \in A \mid x_0 = x)$$
$$\geq \mathbb{P}(x_{T+1} \in A \mid x_1 \in B, x_0 = x) \cdot \mathbb{P}(x_1 \in B \mid x_0 = x) \geq \epsilon,$$

where

$$\epsilon = \delta \cdot \inf_{y \in \mathcal{C}} \mathbb{P}(x_1 \in B \mid x_0 = y) > 0,$$

which then leads to

$$Q(x, A) := \mathbb{P}(x_t \in A, \text{ infinitely often}) = +\infty.$$

This verifies that the chain $\{x_t\}$ is Harris recurrent (see Section 9 in [23] for definition). It can be further derived that for any $x \in \mathcal{C}$,

$$\mathbb{E}(\tau_A \mid x_0 = x) = \sum_{t=1}^{\infty} \mathbb{P}(\tau_A \geq t \mid x_0 = x) \leq (T+1) \sum_{k=0}^{\infty} \mathbb{P}(\tau_A > (T+1)k \mid x_0 = x)$$

$$\leq (T+1) \sum_{k=1}^{\infty} (1-\epsilon)^k < +\infty.$$

The bound above is uniform for all $x \in \mathcal{C}$ and this implies that $\mathcal{C}$ is a regular set of $\{x_t\}$ (see Section 11 in [23] for definition). Finally, one can apply Theorem 13.0.1 in [23] to conclude that there exists a unique invariant probability measure $\nu_{\mathcal{D}}$ on $\mathcal{B}(\mathcal{C})$ and that the distribution of $x_t$ converges weakly to $\nu_{\mathcal{D}|\mathcal{B}}$ conditioned on $x_0 = x$ for any $x \in \mathcal{C}$. $\square$

# F  Proof of Theorem 3.2

*Theorem* (RU guarantee of PNSGD unlearning, fixed $\mathcal{B}$). Assume $\forall \mathbf{d} \in \mathcal{X}$, $f(x; \mathbf{d})$ is $L$-smooth, $M$-Lipchitz and $m$-strongly convex in $x$. Let the learning and unlearning processes follow Algorithm 1 with $y_0^0 = x_T^0 = \mathcal{M}(\mathcal{D})$. Given any fixed mini-batch sequence $\mathcal{B}$, for any $\alpha > 1$, let $\eta \leq \frac{1}{L}$, the output of the $K^{th}$ unlearning iteration satisfies $(\alpha, \varepsilon)$-RU with $c = 1 - \eta m$, where

$$\varepsilon \leq \frac{\alpha - 1/2}{\alpha - 1} \left( \varepsilon_1(2\alpha) + \varepsilon_2(2\alpha) \right),$$

$$\varepsilon_1(\alpha) = \frac{\alpha(2R)^2}{2\eta\sigma^2} c^{2Tn/b}, \quad \varepsilon_2(\alpha) = \frac{\alpha Z_{\mathcal{B}}^2}{2\eta\sigma^2} c^{2Kn/b},$$

$$Z_{\mathcal{B}} = 2Rc^{Tn/b} + \min\left( \frac{1 - c^{Tn/b}}{1 - c^{n/b}} \frac{2\eta M}{b}, 2R \right).$$

We first introduce an additional definition and a lemma needed for the our analysis.

**Definition F.1** (Shifted Rényi divergence). Let $\mu$ and $\nu$ be distributions defined on a Banach space $(\mathbb{R}^d, \| \cdot \|)$. For parameters $z \geq 0$ and $\alpha \geq 1$, the $z$-shifted Rényi divergence between $\mu$ and $\nu$ is defined as

$$D_\alpha^{(z)}(\mu\|\nu) = \inf_{\mu': W_\infty(\mu,\mu') \leq z} D_\alpha(\mu'\|\nu). \tag{6}$$

**Lemma F.2** (Data-processing inequality for Rényi divergence, Lemma 2.6 in [16]). *For any $\alpha \geq 1$, any (random) map $h : \mathbb{R}^d \mapsto \mathbb{R}^d$ and any distribution $\mu, \nu$ with support on $\mathbb{R}^d$,*

$$D_\alpha(h_\#\mu\|h_\#\nu) \leq D_\alpha(\mu\|\nu), \tag{7}$$

*where $h_\#\mu$ denotes the pushforward operation for a function $h$ and distribution $\mu$.*

**Proposition F.3** (Weak Triangle Inequality of Rényi divergence, Corollary 4 in [19]). *For any $\alpha > 1$, $p, q > 1$ satisfying $1/p + 1/q = 1$ and distributions $P, Q, R$ with the same support:*

$$D_\alpha(P\|R) \leq \frac{\alpha - \frac{1}{p}}{\alpha - 1} D_{p\alpha}(P\|Q) + D_{q(\alpha-1/p)}(Q\|R).$$

*Proof.* Recall that from the sketch of proof of Theorem 3.2, we have defined the six distributions: $\nu_{\mathcal{D}|\mathcal{B}}, \nu_{T|\mathcal{B}}^0, \rho_{K|\mathcal{B}}^0$ are the stationary distribution of the learning process, distribution at epoch $T$ of the learning process and distribution at epoch $K$ of the unlearning process. Note that we learn on dataset $\mathcal{D}$ and fine-tune on $\mathcal{D}'$. On the other hand, the corresponding distributions of "adjacent" processes that learn on $\mathcal{D}'$ and still unlearn on $\mathcal{D}'$ are denoted as $\nu_{\mathcal{D}'|\mathcal{B}}, \nu_{T|\mathcal{B}}^{0,\prime}, \rho_{K|\mathcal{B}}^{0,\prime}$ similarly. Note that $\nu_{T|\mathcal{B}}^{0,\prime}$ is the distribution of retraining from scratch on $\mathcal{D}'$, and we aim to bound $d_\alpha(\rho_{K|\mathcal{B}}^0, \nu_{T|\mathcal{B}}^{0,\prime})$ for all possible $\mathcal{D}, \mathcal{D}'$ pairs. By Proposition F.3, we know that for any $\alpha > 1$, by choosing $p = q = 2$, we have

$$d_\alpha(\rho_{K|\mathcal{B}}^0, \nu_{T|\mathcal{B}}^{0,\prime}) \leq \frac{\alpha - 1/2}{\alpha - 1} \left( d_{2\alpha}(\nu_{\mathcal{D}'|\mathcal{B}}, \nu_{T|\mathcal{B}}^{0,\prime}) + d_{2\alpha}(\nu_{\mathcal{D}'|\mathcal{B}}, \rho_{K|\mathcal{B}}^0) \right). \tag{8}$$

Recall the Definition 2.1 that $d_\alpha(P, Q) = \max(D_\alpha(P||Q), D_\alpha(Q||P))$ for distributions $P, Q$. The above inequality is correct since consider any distributions $P, Q, R$, by Proposition F.3 we have

$$D_\alpha(P||R) \leq \frac{\alpha - \frac{1}{2}}{\alpha - 1} D_{2\alpha}(P||Q) + D_{2\alpha - 1}(Q||R) \tag{9}$$

$$\overset{(a)}{\leq} \frac{\alpha - \frac{1}{2}}{\alpha - 1} d_{2\alpha}(P, Q) + d_{2\alpha - 1}(Q, R) \tag{10}$$

$$\overset{(b)}{\leq} \frac{\alpha - \frac{1}{2}}{\alpha - 1} d_{2\alpha}(P, Q) + d_{2\alpha}(Q, R) \tag{11}$$

$$\overset{(c)}{\leq} \frac{\alpha - \frac{1}{2}}{\alpha - 1} \left( d_{2\alpha}(P, Q) + d_{2\alpha}(Q, R) \right) \tag{12}$$

$$\tag{13}$$

where (a) is due to the definition of Rényi difference, (b) is due to the monotonicity of Rényi divergence in $\alpha$, and (c) is due to the fact that for all $\alpha > 1$, $\frac{\alpha - \frac{1}{2}}{\alpha - 1} \geq 1$.

Similarly, we have

$$D_\alpha(R||P) \leq \frac{\alpha - \frac{1}{2}}{\alpha - 1} \left( d_{2\alpha}(P, Q) + d_{2\alpha}(Q, R) \right). \tag{14}$$

Combining these two bounds leads to the weak triangle inequality for Rényi difference.

Now we establish the upper bound of $d_\alpha(\nu_{\mathcal{D}'|\mathcal{B}}, \nu_{T|\mathcal{B}}^{0,'})$, which we denoted as $\varepsilon_1(\alpha)$ in Theorem 3.2. We first note that due to the projection operator $\Pi_{\mathcal{C}_R}$, we trivially have $W_\infty(\nu_{\mathcal{D}'|\mathcal{B}}, \nu_0^0) \leq 2R$. On the other hand, note that $\nu_{T|\mathcal{B}}^{0,'}$ is the distribution of the learning process at epoch $T$ with respect to dataset $\mathcal{D}'$, where $\nu_{\mathcal{D}'|\mathcal{B}}$ is the corresponding stationary distribution. Let us "unroll" iterations so that $v = t\frac{n}{b} + j$. We apply Lemma 2.6 for these two processes, where the initial distribution are $\mu_0 = \nu_0^0$ and $\mu_0' = \nu_{\mathcal{D}'|\mathcal{B}}$. The updates $\psi_v(x) = \Pi_{\mathcal{C}}(x) - \frac{\eta}{b} \sum_{i \in \mathbf{B}^j} \nabla f(\Pi_{\mathcal{C}}(x); \mathbf{d}_i')$ are with respect to the dataset $\mathcal{D}'$, and $\zeta_v = \mathcal{N}(0, \eta\sigma^2 I_d)$. First, note that we have $\eta\sigma^2$ instead of $\sigma^2$ as in the Lemma 2.6, hence we need to apply a change of variable. The only thing we are left to prove is that $\psi_v$ is $c$-contractive for $c = 1 - \eta m$ and any $t, j$. This is because for any mini-batch $\mathcal{B}^j$ and any data point $\mathbf{d}_i' \in \mathcal{X}, i \in \mathcal{B}^j$ of size $b$, we have

$$\|\psi_v(x) - \psi_v(x')\| = \|\Pi_{\mathcal{C}}(x) - \frac{\eta}{b} \sum_{i \in \mathbf{B}^j} \nabla f(\Pi_{\mathcal{C}}(x); \mathbf{d}_i') - \Pi_{\mathcal{C}}(x') + \frac{\eta}{b} \sum_{i \in \mathbf{B}^j} \nabla f(\Pi_{\mathcal{C}}(x'); \mathbf{d}_i')\| \tag{15}$$

$$\leq \frac{1}{b} \sum_{i \in \mathbf{B}^j} \| (\Pi_{\mathcal{C}}(x) - \eta\nabla f(\Pi_{\mathcal{C}}(x); \mathbf{d}_i')) - (\Pi_{\mathcal{C}}(x') - \eta\nabla f(\Pi_{\mathcal{C}}(x'); \mathbf{d}_i')) \| \tag{16}$$

$$\overset{(a)}{\leq} (1 - \eta m)\|\Pi_{\mathcal{C}}(x) - \Pi_{\mathcal{C}}(x')\| \tag{17}$$

$$\leq (1 - \eta m)\|x - x'\|, \tag{18}$$

Finally, due to data-processing inequality for Rényi divergence (Lemma F.2), applying the final projection step does not increase the corresponding Rényi divergence. As a result, by Lemma 2.6 we have

$$D_\alpha(\nu_{\mathcal{D}'|\mathcal{B}}||\nu_{T|\mathcal{B}}^{0,'}) \leq \frac{\alpha W_\infty(\nu_{\mathcal{D}'|\mathcal{B}}, \nu_{T|\mathcal{B}}^{0,'})^2}{2\eta\sigma^2} c^{2Tn/b} \leq \frac{\alpha (2R)^2}{2\eta\sigma^2} c^{2Tn/b}, \tag{19}$$

where the last inequality is due to our naive bound on $W_\infty(\nu_{\mathcal{D}'|\mathcal{B}}, \nu_0^0)$. One can repeat the same analysis for the direction $D_\alpha(\nu_{T|\mathcal{B}}^{0,'}||\nu_{\mathcal{D}'|\mathcal{B}})$, which leads to the same bound by symmetry of Lemma 2.6. Together we have shown that

$$d_\alpha(\nu_{\mathcal{D}'|\mathcal{B}}||\nu_{T|\mathcal{B}}^{0,'}) \leq \frac{\alpha (2R)^2}{2\eta\sigma^2} c^{2Tn/b} = \varepsilon_1(\alpha). \tag{20}$$

Now we focus on bounding the term $d_\alpha(\nu_{\mathcal{D}'|\mathcal{B}}, \rho^0_{K|\mathcal{B}})$. We once again note that $\nu_{\mathcal{D}'|\mathcal{B}}$ is the stationary distribution of the process $\rho^0_{K|\mathcal{B}}$, since we fine-tune with respect to the dataset $\mathcal{D}'$ for the unlearning process. As a result, the same analysis above can be applied, where the only difference is the initial $W_\infty$ distance between $\nu_{\mathcal{D}'|\mathcal{B}}, \rho^0_{0|\mathcal{B}}$.

$$d_\alpha(\nu_{\mathcal{D}'|\mathcal{B}}, \rho^0_{K|\mathcal{B}}) \le \frac{\alpha W_\infty(\nu_{\mathcal{D}'|\mathcal{B}}, \rho^0_{0|\mathcal{B}})^2}{2\eta\sigma^2} c^{2Kn/b}. \tag{21}$$

We are left with establish an upper bound of $W_\infty(\nu_{\mathcal{D}'|\mathcal{B}}, \rho^0_{0|\mathcal{B}})$. Note that since $W_\infty$ is indeed a metric, we can apply triangle inequality which leads to the following upper bound.

$$W_\infty(\nu_{\mathcal{D}'|\mathcal{B}}, \rho^0_{0|\mathcal{B}}) \le W_\infty(\nu_{\mathcal{D}'|\mathcal{B}}, \nu_{\mathcal{D}|\mathcal{B}}) + W_\infty(\nu_{\mathcal{D}|\mathcal{B}}, \rho^0_{0|\mathcal{B}}) \tag{22}$$

From Lemma 3.3, we have that

$$W_\infty(\nu^0_{T|\mathcal{B}}, \nu^{0,\prime}_{T|\mathcal{B}}) \le \min\left(\frac{1 - c^{Tn/b}}{1 - c^{n/b}} c^{n/b - j_0 - 1} \frac{2\eta M}{b}, 2R\right) \le \min\left(\frac{1}{1 - c^{n/b}} \frac{2\eta M}{b}, 2R\right), \tag{23}$$

where the last inequality is simply due to the fact that $c < 1$. Since the upper bound is independent of $T$, by letting $T \to \infty$, and the definition that $\nu_{\mathcal{D}|\mathcal{B}}, \nu_{\mathcal{D}'|\mathcal{B}}$ are the limiting distribution of $\nu^0_{T|\mathcal{B}}, \nu^{0,\prime}_{T|\mathcal{B}}$ respectively, we have

$$W_\infty(\nu_{\mathcal{D}|\mathcal{B}}, \nu_{\mathcal{D}'|\mathcal{B}}) \le \min\left(\frac{1}{1 - c^{n/b}} \frac{2\eta M}{b}, 2R\right). \tag{24}$$

On the other hand, note that by definition we know that $\rho^0_{0|\mathcal{B}} = \nu^0_{T|\mathcal{B}}$. Thus by Lemma 3.4 we have (recall that $c = 1 - \eta m$)

$$W_\infty(\nu_{\mathcal{D}|\mathcal{B}}, \rho^0_{0|\mathcal{B}}) = W_\infty(\nu^0_{T|\mathcal{B}}, \nu_{\mathcal{D}|\mathcal{B}}) \le c^{Tn/b} W_\infty(\nu^0_0, \nu_{\mathcal{D}|\mathcal{B}}) \le 2R \times c^{Tn/b}, \tag{25}$$

where the last inequality is again due to the naive bound induced by the projection to $\mathcal{C}_R$ for $W_\infty(\nu^0_0, \nu_{\mathcal{D}|\mathcal{B}})$. Together we have that

$$W_\infty(\nu_{\mathcal{D}'|\mathcal{B}}, \rho^0_{0|\mathcal{B}}) \le 2R \times c^{Tn/b} + \min\left(\frac{1}{1 - c^{n/b}} \frac{2\eta M}{b}, 2R\right) = Z_\mathcal{B}. \tag{26}$$

Combining things we complete the proof. $\qquad\square$

## G   Proof of Theorem 3.11

*Theorem.* Under the same assumptions as in Corollary 3.7. Assume the unlearning requests arrive sequentially such that our dataset changes from $\mathcal{D} = \mathcal{D}_0 \to \mathcal{D}_1 \to \ldots \to \mathcal{D}_S$, where $\mathcal{D}_s, \mathcal{D}_{s+1}$ are adjacent. Let $y^{j,(s)}_k$ be the unlearned parameters for the $s^{th}$ unlearning request at $k^{th}$ unlearning epoch and $j^{th}$ iteration following Algorithm (1) on $\mathcal{D}_s$ and $y^{0,(s+1)}_0 = y^{0,(s)}_{K_s} \sim \bar\nu_{\mathcal{D}_s|\mathcal{B}}$, where $y^{0,(0)}_0 = x_\infty$ and $K_s$ is the unlearning steps for the $s^{th}$ unlearning request. For any $s \in [S]$, we have

$$W_\infty(\bar\nu_{\mathcal{D}_{s-1}|\mathcal{B}}, \nu_{\mathcal{D}_s|\mathcal{B}}) \le Z^{(s)}_\mathcal{B},$$

where $Z^{(s+1)}_\mathcal{B} = \min(c^{K_s n/b} Z^{(s)}_\mathcal{B} + Z_\mathcal{B}, 2R)$, $Z^{(1)}_\mathcal{B} = Z_\mathcal{B}$, $Z_\mathcal{B}$ is described in Corollary 3.7 and $c = 1 - \eta m$.

The proof is a direct application of triangle inequality, Lemma 3.4 and 3.3. We will prove it by induction. For the base case $s = 1$ it trivial, since $\bar\nu_{\mathcal{D}_0} = \nu_{\mathcal{D}_0}$ as we choose $y^{0,(0)}_0 = x^0_\infty$. Thus by our definition that $Z_\mathcal{B}$ is the upper bound of $W_\infty(\nu_{\mathcal{D}|\mathcal{B}}, \nu_{\mathcal{D}'|\mathcal{B}})$ for any adjacent dataset $\mathcal{D}, \mathcal{D}'$. Apparently, we also have

$$W_\infty(\bar\nu_{\mathcal{D}_0|\mathcal{B}}, \nu_{\mathcal{D}_1|\mathcal{B}}) = W_\infty(\nu_{\mathcal{D}_0|\mathcal{B}}, \nu_{\mathcal{D}_1|\mathcal{B}}) \le Z_\mathcal{B} = Z^{(1)}_\mathcal{B} \tag{27}$$

For the induction step, suppose our hypothesis is true until $s$ step. Then for the $s + 1$ step we have

$$W_\infty(\bar{\nu}_{\mathcal{D}_s|\mathcal{B}}, \nu_{\mathcal{D}_{s+1}|\mathcal{B}}) \overset{(a)}{\leq} W_\infty(\bar{\nu}_{\mathcal{D}_s|\mathcal{B}}, \nu_{\mathcal{D}_s|\mathcal{B}}) + W_\infty(\nu_{\mathcal{D}_s|\mathcal{B}}, \nu_{\mathcal{D}_{s+1}|\mathcal{B}}) \tag{28}$$

$$\overset{(b)}{\leq} W_\infty(\bar{\nu}_{\mathcal{D}_s|\mathcal{B}}, \nu_{\mathcal{D}_s|\mathcal{B}}) + Z_{\mathcal{B}} \tag{29}$$

$$\overset{(c)}{\leq} c^{K_{s-1}n/b} W_\infty(\bar{\nu}_{\mathcal{D}_{s-1}|\mathcal{B}}, \nu_{\mathcal{D}_s|\mathcal{B}}) + Z_{\mathcal{B}} \tag{30}$$

$$\overset{(d)}{\leq} c^{K_{s-1}n/b} Z_{\mathcal{B}}^{(s)} + Z_{\mathcal{B}} \tag{31}$$

where $(a)$ is due to triangle inequality as $W_\infty$ is a metric. $(b)$ is due to Corollary 3.7, where $Z_{\mathcal{B}}$ is an upper bound of $W_\infty$ distance between any two adjacent stationary distributions. $(c)$ is due to Lemma 3.4 and $(d)$ is due to our hypothesis. Finally, note that $2R$ is a natural universal upper bound due to our projection on $\mathcal{C}_R$, which has diameter $2R$. Together we complete the proof.

# H    Proof of Corollary 3.7

Note that under the training convergent assumption, the target retraining distribution is directly $\nu_{\mathcal{D}'|\mathcal{B}}$ so that we do not need the weak triangle inequality for Rényi difference. Similarly, we do not need the triangle inequality for the term $Z_{\mathcal{B}}$. Directly using $\varepsilon_2(\alpha)$ with $Z_{\mathcal{B}} = \min\left(\frac{1}{1-c^{n/b}}\frac{2\eta M}{b}, 2R\right)$ from Theorem 3.2 leads to the result.

# I    Improved bound with randomized $\mathcal{B}$

*Corollary.* Under the same setting as of Theorem 3.2 but with random mini-batch sequences described in Algorithm 1. Then for any $\alpha > 1$, and $\eta \leq \frac{1}{L}$, the output of the $K^{th}$ unlearning iteration satisfies $(\alpha, \varepsilon)$-RU with $c = 1 - \eta m$, where

$$\varepsilon \leq \frac{1}{\alpha - 1} \log\left(\mathbb{E}_{\mathcal{B}} \exp\left(\frac{\alpha(\alpha - 1)Z_{\mathcal{B}}^2}{2\eta\sigma^2} c^{2Kn/b}\right)\right),$$

where $Z_{\mathcal{B}}$ is the bound described in Lemma 3.3.

We first restate the lemma in [22], which is an application of the joint convexity of KL divergence.

**Lemma I.1** (Lemma 4.1 in [22])**.** *Let $\nu_1, \cdots, \nu_m$ and $\nu'_1, \cdots, \nu'_m$ be distributions over $\mathbb{R}^d$. For any $\alpha \geq 1$ and any coefficients $p_1, \cdots, p_m \geq 0$ such that $\sum_{i=1}^m p_i = 1$, the following inequality holds.*

$$\exp((\alpha - 1)D_\alpha(\sum_{i=1}^m p_i\nu_i || \sum_{i=1}^m p_i\nu'_i)) \tag{32}$$

$$\leq \sum_{i=1}^m p_i \exp((\alpha - 1)D_\alpha(\nu_i||\nu'_i)). \tag{33}$$

The proof of Corollary 3.8 directly follows by repeating the proof of Lemma 3.7 but with the $\mathcal{B}$ dependent $Z_{\mathcal{B}}$ described in Lemma 3.3. Then leverage Lemma I.1 leads to the result.

# J    $W_\infty$ bound for batch unlearning

**Corollary J.1** ($W_\infty$ bound for batch unlearning)**.** *Consider the learning process in Algorithm 1 on adjacent dataset $\mathcal{D}$ and $\mathcal{D}'$ that differ in $1 \leq S$ points and a fixed mini-batch sequence $\mathcal{B}$. Assume $\forall \mathbf{d} \in \mathcal{X}$, $f(x; \mathbf{d})$ is $L$-smooth, $M$-Lipschitz and $m$-strongly convex in $x$. Let the index of different data point between $\mathcal{D}, \mathcal{D}'$ belongs to mini-batches $\mathcal{B}^{j_0}, \mathcal{B}^{j_1}, \ldots, \mathcal{B}^{j_{G-1}}$, each of which contains $1 \leq S_{j_g} \leq b$ such that $\sum_{g=0}^{G-1} S_{j_g} = S$. Then for $\eta \leq \frac{1}{L}$ and $c = (1 - \eta m)$, we have*

$$W_\infty(\nu_{T|\mathcal{B}}^0, \nu_{T|\mathcal{B}}^{0,'}) \leq \min\left(\frac{1 - c^T}{1 - c} \sum_{g=0}^{G-1} c^{n/b - j_g - 1}\frac{2\eta M S_{j_g}}{b}, 2R\right).$$

*Proof.* The proof is a direct generalization of Lemma 3.3. Recall that for the per-iteration bound within an epoch, there are two possible cases: 1) we encounter the mini-batch $\mathcal{B}^j$ that contains indices of replaced points 2) or not. This is equivalently to 1) $j \in \{j_g\}_{g=0}^{G-1}$ or 2) $j \notin \{j_g\}_{g=0}^{G-1}$.

Let us assume that case 1) happens and $j = j_g$ for $g \in \{0, \dots, G-1\}$. Also, let us denote the set of those indices as $S_{j_g}$ with a slight abuse of notation. By the same analysis, we know that

$$\|\psi_j(x_t^j) - \psi_j'(x_t^{j,\prime})\| \tag{34}$$

$$\leq \frac{1}{b} \sum_{i \in \mathcal{B}^j \setminus S_{j_g}} \|x_t^j - x_t^{j,\prime} + \eta(\nabla f(x_t^j; \mathbf{d}_i) - \nabla f(x_t^{j,\prime}; \mathbf{d}_i))\| \tag{35}$$

$$+ \frac{1}{b} \sum_{i \in S_{j_g}} \|x_t^j - x_t^{j,\prime} + \eta(\nabla f(x_t^j; \mathbf{d}_i) - \nabla f(x_t^{j,\prime}; \mathbf{d}_i'))\|. \tag{36}$$

For the first term, note that the gradient mapping (with respect to the same data point) is contractive with constant $(1 - \eta m)$ for $\eta \leq \frac{1}{L}$, thus we have

$$\|x_t^j - x_t^{j,\prime} + \eta(\nabla f(x_t^j; \mathbf{d}_i) - \nabla f(x_t^{j,\prime}; \mathbf{d}_i))\| \leq (1 - \eta m)\|x_t^j - x_t^{j,\prime}\|. \tag{37}$$

For the second term, by triangle inequality and the $M$-Lipschitzness of $f$ we have

$$\|x_t^j - x_t^{j,\prime} + \eta(\nabla f(x_t^j; \mathbf{d}_i) - \nabla f(x_t^{j,\prime}; \mathbf{d}_i'))\| \tag{38}$$

$$= \|x_t^j - x_t^{j,\prime} + \eta(\nabla f(x_t^j; \mathbf{d}_i) - \nabla f(x_t^{j,\prime}; \mathbf{d}_i) + \nabla f(x_t^{j,\prime}; \mathbf{d}_i) - \nabla f(x_t^{j,\prime}; \mathbf{d}_i'))\| \tag{39}$$

$$\leq (1 - \eta m)\|x_t^j - x_t^{j,\prime}\| + \eta\|\nabla f(x_t^{j,\prime}; \mathbf{d}_i)\| + \eta\|\nabla f(x_t^{j,\prime}; \mathbf{d}_i')\| \tag{40}$$

$$\leq (1 - \eta m)\|x_t^j - x_t^{j,\prime}\| + 2\eta M. \tag{41}$$

Note that there are $S_{j_g}$ terms above. Combining things together, we have

$$\|\psi_j(x_t^j) - \psi_j'(x_t^{j,\prime})\| \leq (1 - \eta m)\|x_t^j - x_t^{j,\prime}\| + \frac{2\eta M S_{j_g}}{b}. \tag{42}$$

Iterate this bound over all $n/b$ iterations within this epoch, we have

$$\|\psi_j(x_{t+1}^0) - \psi_j'(x_{t+1}^{0,\prime})\| \leq (1 - \eta m)^{n/b}\|\psi_j(x_t^0) - \psi_j'(x_t^{0,\prime})\| + \sum_{g=0}^{G-1}(1 - \eta m)^{n/b - j_g - 1}\frac{2\eta M S_{j_g}}{b}. \tag{43}$$

The rest analysis is the same as those in Lemma 3.3, where we iterate over $T$ epochs, infimum over all possible coupling, and then simplify the expression using geometric series. Thus we complete the proof.

$\square$

## K   Proof of Proposition 3.9

*Proposition.* Under the same setting as Corollary 3.7 with $b = n$, $\eta \leq \frac{m}{2L^2}$ and assume the minimizer of any $f_\mathcal{D}$ is in the relative interior of $\mathcal{C}_R \subseteq \mathbb{R}^d$, for any given adjacent dataset pair $\mathcal{D}, \mathcal{D}'$ the output of the $K^{th}$ unlearning iteration $y_K^0$ satisfies

$$\mathbb{E}[f_{\mathcal{D}'}(y_K^0) - \inf_{x \in \mathcal{C}_R} f_{\mathcal{D}'}(x)] \leq Mc^K \min(\frac{1}{1-c}\frac{2\eta M}{n}, 2R) + \frac{2Ld\sigma^2}{m}. \tag{44}$$

The proof is based on the following key lemma from [24].

**Lemma K.1.** *For $M$-Lipschitz, $m$-strongly convex and $L$-smooth loss function $f_\mathcal{D}(x)$ over the closed bounded convex set $\mathcal{C}_R \subseteq \mathbb{R}^d$ with radius $R$, step size $\eta \leq \frac{m}{2L^2}$ and initial parameter $x_0 \sim \nu_0$, the excess empirical risk of the learning process of Algorithm 1 for $T$ epoch is bounded by*

$$\mathbb{E}[f_\mathcal{D}(x_T^0) - f_\mathcal{D}(x^\star)] \leq \frac{L}{2}\exp(-m\eta T)\mathbb{E}_{x_0 \sim \nu_0}[\|x_0 - x^\star\|^2] + \frac{2Ld\sigma^2}{m}, \tag{45}$$

*where $x^\star$ is the minimizer of $f_\mathcal{D}(x)$ in the relative interior of $\mathcal{C}_R$ and $d$ is the dimension of parameter.*

Clearly, we can see that under the convergent assumption of Corollary 3.7, the excess risk of $y_0^0 = x_\infty$ is

$$\mathbb{E}[f_\mathcal{D}(y_0^0) - f_\mathcal{D}(x^\star)] \leq \frac{2Ld\sigma^2}{m}. \tag{46}$$

This is true since $\|x_0 - x^\star\| \leq 2R$ is finite by the boundedness of $\mathcal{C}_R$.

Now we are ready to prove Proposition 3.9.

*Proof.* Start by observing that

$$\mathbb{E}[f_{\mathcal{D}'}(y_K^0) - f_{\mathcal{D}'}(y^\star)] = \mathbb{E}[f_{\mathcal{D}'}(y_K^0) - f_{\mathcal{D}'}(y_K^{0,'})] + \mathbb{E}[f_{\mathcal{D}'}(y_K^{0,'}) - f_{\mathcal{D}'}(y^\star)], \tag{47}$$

where we recall that $y_K^{0,'}$ is the "adjacent process" of $y_K^0$ that running noisy gradient descent on $\mathcal{D}'$ for both learning and unlearning processes. As a result, we have $y_K^{0,'} = x_{T+K}^{0,'}$ for $T$ learning epoch. By taking $T \to \infty$ under the convergent assumption and utilizing Lemma K.1, we have

$$\mathbb{E}[f_{\mathcal{D}'}(y_K^{0,'}) - f_{\mathcal{D}'}(y^\star)] \leq \frac{2Ld\sigma^2}{m}. \tag{48}$$

Hence we are left with analyzing the first term. We will bound it in terms of $\|y_K^0 - y_K^{0,'}\|$. Note that we choose a particular coupling between $y_K^0, y_K^{0,'}$ as in Lemma 3.3, where the Gaussian noise in $y_K^0, y_K^{0,'}$ are identical almost surely. Also for simplicity, we drop 0 in the superscript since we are analyzing the full batch case. By $M$-Lipschitzness of $f_{\mathcal{D}'}$, we have

$$\|f_{\mathcal{D}'}(y_K) - f_{\mathcal{D}'}(y_K')\| \leq M\|y_K^0 - y_K^{0,'}\|. \tag{49}$$

For $\|y_K^0 - y_K^{0,'}\|$, we have

$$\|y_K - y_K'\| \tag{50}$$
$$= \|\Pi_{\mathcal{C}_R}[y_{K-1} - \eta\nabla f_{\mathcal{D}'}(y_{K-1}) + \sqrt{2\eta\sigma^2}W_{K-1}] - \Pi_{\mathcal{C}_R}[y_{K-1}' - \eta\nabla f_{\mathcal{D}'}(y_{K-1}') + \sqrt{2\eta\sigma^2}W_{K-1}]\| \tag{51}$$
$$\leq \|y_{K-1} - \eta\nabla f_{\mathcal{D}'}(y_{K-1}) - (y_{K-1}' - \eta\nabla f_{\mathcal{D}'}(y_{K-1}'))\| \tag{52}$$
$$\leq (1 - \eta m)\|y_{K-1} - y_{K-1}'\|, \tag{53}$$

where the first inequality is due to the fact that $\Pi_{\mathcal{C}_R}$ is a contraction map and noise cancels out due to our coupling of $y_K^0, y_K^{0,'}$. The second inequality is due to the fact that gradient update is $(1 - m\eta)$-contraction map when $f_{\mathcal{D}'}$ is $L$-smooth, $m$-strongly convex and $\eta \leq \frac{1}{L}$. By iterating this bound, we have

$$\|f_{\mathcal{D}'}(y_K) - f_{\mathcal{D}'}(y_K')\| \leq M(1 - m\eta)^K\|y_0 - y_0'\| \tag{54}$$
$$\leq Mc^K \min(\frac{1}{1-c}\frac{2\eta M}{n}, 2R), \tag{55}$$

where $c = 1 - m\eta$ and the last inequality is due to Lemma 3.3 for the case $b = n$ and taking the limit $T \to \infty$. Combining all results together, we have

$$\mathbb{E}[f_{\mathcal{D}'}(y_K^0) - f_{\mathcal{D}'}(y^\star)] \tag{56}$$
$$\leq \mathbb{E}[f_{\mathcal{D}'}(y_K^0) - f_{\mathcal{D}'}(y_K^{0,'})] + \frac{2Ld\sigma^2}{m} \tag{57}$$
$$\leq \mathbb{E}[\|f_{\mathcal{D}'}(y_K^0) - f_{\mathcal{D}'}(y_K^{0,'})\|] + \frac{2Ld\sigma^2}{m} \tag{58}$$
$$\leq Mc^K \min(\frac{1}{1-c}\frac{2\eta M}{n}, 2R) + \frac{2Ld\sigma^2}{m}. \tag{59}$$

Together we complete the proof. $\qquad\square$

# L   Proof of Lemma 3.3

*Lemma* ($W_\infty$ between adjacent PNSGD learning processes). Consider the learning process in Algorithm 1 on adjacent dataset $\mathcal{D}$ and $\mathcal{D}'$ and a fixed mini-batch sequence $\mathcal{B}$. Assume $\forall \mathbf{d} \in \mathcal{X}$, $f(x; \mathbf{d})$ is $L$-smooth, $M$-Lipschitz and $m$-strongly convex in $x$. Let the index of different data point between $\mathcal{D}, \mathcal{D}'$ belongs to mini-batch $\mathcal{B}^{j_0}$. Then for $\eta \leq \frac{1}{L}$ and let $c = (1 - \eta m)$, we have

$$W_\infty(\nu^0_{T|\mathcal{B}}, \nu^{0,'}_{T|\mathcal{B}}) \leq \min\left(\frac{1 - c^{Tn/b}}{1 - c^{n/b}} c^{n/b - j_0 - 1} \frac{2\eta M}{b}, 2R\right).$$

*Proof.* We prove the bound of one iteration and iterate over that result under a specific coupling of the adjacent PNSGD learning processes, which is by definition an upper bound for taking infimum over all possible coupling. Given an mini-batch sequence $\mathcal{B}$, assume the adjacent dataset $\mathcal{D}, \mathcal{D}'$ differ at index $i_0 \in [n]$ and $i_0$ belongs to $j_0^{th}$ mini-batch (i.e., $i_0 \in \mathcal{B}^{j_0}$. For simplicity, we drop the condition on $\mathcal{B}$ for all quantities in the proof as long as it is clear that we always condition on a fixed $\mathcal{B}$. Let us denote $\psi_j(x) = x - \frac{\eta}{b} \sum_{i \in \mathbf{B}^j} \nabla f(x; \mathbf{d}'_i)$ and similar for $\psi'_j$ on $\mathcal{D}'$. Note that for some coupling of $\nu^j_t, \nu^{j,'}_t$ denoted as $\gamma^j_t$,

$$\text{esssup}_{(x^{j+1}_t, x^{j+1,'}_t) \sim \gamma^{j+1}_t} \|x^{j+1}_t - x^{j+1,'}_t\| \tag{60}$$

$$= \text{esssup}_{((x^j_t, W^j_t), (x^{j,'}_t, W^{j,'}_t)) \sim \gamma^j_t} \|\psi_j(x^j_t) - \psi'_j(x^{j,'}_t) + \sqrt{2\eta\sigma^2} W^j_t - \sqrt{2\eta\sigma^2} W^{j,'}_t\| \tag{61}$$

$$\leq \text{esssup}_{((x^j_t, W^j_t), (x^{j,'}_t, W^{j,'}_t)) \sim \gamma^j_t} \|\psi_j(x^j_t) - \psi'_j(x^{j,'}_t)\| + \|\sqrt{2\eta\sigma^2} W^j_t - \sqrt{2\eta\sigma^2} W^{j,'}_t\|. \tag{62}$$

Now, note that by definition the noise $W^j_t$ is independent of $x^j_t$, we can simply choose the coupling $\gamma$ so that $W^j_t = W^{j,'}_t$ for all $t, j$ and $x^0_0 = x^{0,'}_0$ due to the same initialization distribution. So the last term is 0. For the first term, there are two possible cases: 1) $j = j_0$ and 2) $j \neq j_0$. For case 1), we have

$$\|\psi_j(x^j_t) - \psi'_j(x^{j,'}_t)\| \tag{63}$$

$$\leq \frac{1}{b} \sum_{i \in \mathcal{B}^j \setminus \{i_0\}} \|x^j_t - x^{j,'}_t + \eta(\nabla f(x^j_t; \mathbf{d}_i) - \nabla f(x^{j,'}_t; \mathbf{d}_i))\| \tag{64}$$

$$+ \frac{1}{b} \|x^j_t - x^{j,'}_t + \eta(\nabla f(x^j_t; \mathbf{d}_{i_0}) - \nabla f(x^{j,'}_t; \mathbf{d}'_{i_0}))\|. \tag{65}$$

For the first term, note that the gradient mapping (with respect to the same data point) is contractive with constant $(1 - \eta m)$ for $\eta \leq \frac{1}{L}$, thus we have

$$\|x^j_t - x^{j,'}_t + \eta(\nabla f(x^j_t; \mathbf{d}_i) - \nabla f(x^{j,'}_t; \mathbf{d}_i))\| \leq (1 - \eta m)\|x^j_t - x^{j,'}_t\|. \tag{66}$$

For the second term, by triangle inequality and the $M$-Lipschitzness of $f$ we have

$$\|x^j_t - x^{j,'}_t + \eta(\nabla f(x^j_t; \mathbf{d}_{i_0}) - \nabla f(x^{j,'}_t; \mathbf{d}'_{i_0}))\| \tag{67}$$

$$= \|x^j_t - x^{j,'}_t + \eta(\nabla f(x^j_t; \mathbf{d}_{i_0}) - \nabla f(x^{j,'}_t; \mathbf{d}_{i_0}) + \nabla f(x^{j,'}_t; \mathbf{d}_{i_0}) - \nabla f(x^{j,'}_t; \mathbf{d}'_{i_0}))\| \tag{68}$$

$$\leq (1 - \eta m)\|x^j_t - x^{j,'}_t\| + \eta\|\nabla f(x^{j,'}_t; \mathbf{d}_{i_0})\| + \eta\|\nabla f(x^{j,'}_t; \mathbf{d}'_{i_0})\| \tag{69}$$

$$\leq (1 - \eta m)\|x^j_t - x^{j,'}_t\| + 2\eta M. \tag{70}$$

Combining things together, we have

$$\|\psi_j(x^j_t) - \psi'_j(x^{j,'}_t)\| \leq (1 - \eta m)\|x^j_t - x^{j,'}_t\| + \frac{2\eta M}{b}. \tag{71}$$

On the other hand, for case 2) we can simply use the contrativity of the gradient update for all $b$ terms, which leads to

$$\|\psi_j(x^j_t) - \psi'_j(x^{j,'}_t)\| \leq (1 - \eta m)\|x^j_t - x^{j,'}_t\|. \tag{72}$$

Combining these two cases, we have

$$\text{esssup}_\gamma \|x^{j+1}_t - x^{j+1,'}_t\| \leq \text{esssup}_\gamma (1 - \eta m)\|x^j_t - x^{j,'}_t\| + \frac{2\eta M}{b} \mathbf{1}[j = j_0] \tag{73}$$

$$\tag{74}$$

where $\mathbf{1}[j = j_0]$ is the indicator function of the event $j = j_0$. Now we iterate this bound within the epoch $t$, which leads to an epoch-wise bound

$$\text{esssup}_\gamma \|x_{t+1}^0 - x_{t+1}^{0,'}\| \leq \text{esssup}_\gamma (1 - \eta m)^{n/b} \|x_t^0 - x_t^{0,'}\| + (1 - \eta m)^{n/b - j_0 - 1} \frac{2\eta M}{b} \quad (75)$$

$$(76)$$

We can further iterate this bound over all iterations $T$, which leads to

$$\text{esssup}_\gamma \|x_T^0 - x_T^{0,'}\| \leq \text{esssup}_\gamma (1 - \eta m)^{Tn/b} \|x_0^0 - x_0^{0,'}\| + \sum_{t=0}^{T-1} (1 - \eta m)^{(t+1)n/b - j_0 - 1} \frac{2\eta M}{b} \quad (77)$$

$$= \sum_{t=0}^{T-1} (1 - \eta m)^{(t+1)n/b - j_0 - 1} \frac{2\eta M}{b}, \quad (78)$$

$$(79)$$

where the last equality follows by our choice of coupling $\gamma$ such that $x_0^0 = x_0^{0,'}$ due to the same initialization distribution. Now by taking the infimum overall possible coupling $\gamma$ and simply the expression, we have

$$W_\infty(\nu_{T|\mathcal{B}}^0, \nu_{T|\mathcal{B}}^{0,'}) \leq \frac{1 - c^{Tn/b}}{1 - c^{n/b}} c^{n/b - j_0 - 1} \frac{2\eta M}{b}. \quad (80)$$

Finally, note that there is also a naive bound $W_\infty(\nu_{T|\mathcal{B}}^0, \nu_{T|\mathcal{B}}^{0,'}) \leq 2R$ due to the projection $\Pi_{\mathcal{C}_R}$. Combine these two we complete the proof. □

## M  Proof of Lemma 3.4

*Lemma ($W_\infty$ between PNSGD learning process to its stationary distribution).* Following the same setting as in Theorem 3.2 and denote the initial distribution of the unlearning process as $\nu_0^0$. Then we have

$$W_\infty(\nu_{T|\mathcal{B}}^0, \nu_{\mathcal{D}|\mathcal{B}}^0) \leq (1 - \eta m)^{Tn/b} W_\infty(\nu_0^0, \nu_{\mathcal{D}|\mathcal{B}}^0).$$

*Proof.* We follow a similar analysis as in Lemma 3.3 but with two key differences: 1) our initial $W_\infty$ distance is not zero and 2) we are applying the same $c$-CNI. Let us slightly abuse the notation to denote the process $x_t^{j,'} \sim \nu_t^{j,'}$ be the process of learning on $\mathcal{D}$ as well but starting with $\nu_0^{0,'} = \nu_{\mathcal{D}|\mathcal{B}}$. As before, we first establish one iteration bound with epoch $t$, then iterate over $n/b$ iterations, and then iterate over $T$ to complete the proof. Consider the two CNI processes $x_t^j, x_t^{j,'}$ with the same update $\psi_j(x) = x - \frac{\eta}{b} \sum_{i \in \mathcal{B}^j} \nabla f(x; \mathbf{d}_i)$, where $x_0^0 \sim \nu_0^0$ and $x_0^{0,'} \sim \nu_{\mathcal{D}|\mathcal{B}}$. We will use the similar coupling construction as in the proof of Lemma 3.3 (i.e., $W_t^j = W_t^{j,'}$). However, note that we will not restrict the coupling between the two initial distributions $\nu_0^0 = \nu_{0|\mathcal{B}}$ and $\nu_0^{0,'} = \nu_{\mathcal{D}|\mathcal{B}}$. Let us denote the coupling for $W$ as $\gamma_W$ and the coupling for the initial distributions $\nu_0^0 = \nu_{0|\mathcal{B}}$ and $\nu_0^{0,'} = \nu_{\mathcal{D}|\mathcal{B}}$ as $\gamma_I$. We denote the overall coupling as $\gamma = (\gamma_I, \gamma_W)$. For our specific choice of coupling $\gamma_W$, we have

$$\|x_t^{j+1} - x_t^{j+1,'}\| \overset{(a)}{\leq} \|\psi_j(x_t^j) - \psi_j(x_t^{j,'}) + \sqrt{2\eta\sigma^2}(W_t^j - W_t^{j,'})\| \overset{(b)}{\leq} \|\psi_j(x_t^j) - \psi_j(x_t^{j,'})\| + 0 \overset{(c)}{\leq} (1 - \eta m)\|x_t^j - x_t^{j,'}\|, \quad (81)$$

where (a) is due to the fact that projection $\Pi_{\mathcal{C}_R}$ is 1-contractive, (b) is due to our coupling choice on the noise distribution, and (c) is due to the fact that $\psi_j$ is $(1 - \eta m)$ contractive and the coupling choice on the mini-batch.

Now we iterate the bound above within the epoch and then over $T$ epoch similar as before, which leads to the per epoch bound as follows

$$\text{esssup}_\gamma \|x_T^0 - x_T^{0,'}\| \leq \text{esssup}_\gamma (1 - \eta m)^{Tn/b} \|x_0^0 - x_0^{0,'}\| = \text{esssup}_{\gamma_I} (1 - \eta m)^{Tn/b} \|x_0^0 - x_0^{0,'}\|. \quad (82)$$

Note that this holds for all coupling $\gamma_I$ for initial distributions. Let us now choose a specific (tie break arbitrary) $\gamma_I$ so that the $W_\infty(\nu_{0|\mathcal{B}}, \nu_{\mathcal{D}|\mathcal{B}}) = \mathrm{esssup}_{\gamma_I}\|x_0^0 - x_0^{0,'}\|$ is attained. For this specific coupling $\gamma_I$, we have

$$\mathrm{esssup}_{(\gamma_I, \gamma_W)}\|x_T^0 - x_T^{0,'}\| \leq (1 - \eta m)^{Tn/b} W_\infty(\nu_{0|\mathcal{B}}, \nu_{\mathcal{D}|\mathcal{B}}). \tag{83}$$

Finally, by the definition of infimum and observe that $x_T^{0,'} \sim \nu_{\mathcal{D}|\mathcal{B}}$ due to the stationary property of $\nu_{\mathcal{D}|\mathcal{B}}$ (which is independent of our choice of coupling), we have

$$W_\infty(\nu_{T|\mathcal{B}}^0, \nu_{\mathcal{D}|\mathcal{B}}) \leq \mathrm{esssup}_{(\gamma_I, \gamma_W)}\|x_T^0 - x_T^{0,'}\| \leq (1 - \eta m)^{Tn/b} W_\infty(\nu_{0|\mathcal{B}}, \nu_{\mathcal{D}|\mathcal{B}}). \tag{84}$$

Hence we complete the proof. $\square$

# N  Experiment Details

## N.1  $(\alpha, \varepsilon)$-RU to $(\epsilon, \delta)$-Unlearning Conversion

Let us first state the definition of $(\epsilon, \delta)$-unlearning from prior literature [7–9].

**Definition N.1.** Consider a randomized learning algorithm $\mathcal{M} : \mathcal{X}^n \mapsto \mathbb{R}^d$ and a randomized unlearning algorithm $\mathcal{U} : \mathbb{R}^d \times \mathcal{X}^n \times \mathcal{X}^n \mapsto \mathbb{R}^d$. We say $(\mathcal{M}, \mathcal{U})$ achieves $(\epsilon, \delta)$-unlearning if for any adjacent datasets $\mathcal{D}, \mathcal{D}'$ and any event $E$, we have

$$\mathbb{P}\left(\mathcal{U}(\mathcal{M}(\mathcal{D}), \mathcal{D}, \mathcal{D}') \subseteq E\right) \leq \exp(\epsilon)\mathbb{P}\left(\mathcal{M}(\mathcal{D}') \subseteq E\right) + \delta, \tag{85}$$

$$\mathbb{P}\left(\mathcal{M}(\mathcal{D}') \subseteq E\right) \leq \exp(\epsilon)\mathbb{P}\left(\mathcal{U}(\mathcal{M}(\mathcal{D}), \mathcal{D}, \mathcal{D}') \subseteq E\right) + \delta. \tag{86}$$

Following the same proof of RDP-DP conversion (Proposition 3 in [19]), we have the following $(\alpha, \varepsilon)$-RU to $(\epsilon, \delta)$-unlearning conversion as well.

**Proposition N.2.** If $(\mathcal{M}, \mathcal{U})$ achieves $(\alpha, \varepsilon)$-RU, it satisfies $(\epsilon, \delta)$-unlearning as well, where

$$\epsilon = \varepsilon + \frac{\log(1/\delta)}{\alpha - 1}. \tag{87}$$

## N.2  Datasets

**MNIST** [25] contains the grey-scale image of number 0 to number 9, each with $28 \times 28$ pixels. We follow [9] to take the images with the label 3 and 8 as the two classes for logistic regression. The training data contains 11264 instances in total and the testing data contains 1984 samples. We spread the image into an $x \in \mathbb{R}^d, d = 724$ feature as the input of logistic regression.

**CIFAR-10** [26] contains the RGB-scale image of ten classes for image classification, each with $32 \times 32$ pixels. We also select class #3 (cat) and class #8 (ship) as the two classes for logistic regression. The training data contains 9728 instances and the testing data contains 2000 samples. We apply data pre-processing on CIFAR-10 by extracting the compact feature encoding from the last layer before pooling of an off-the-shelf pre-trained ResNet18 model [27] from Torch-vision library [34, 35] as the input of our logistic regression. The compact feature encoding is $x \in \mathbb{R}^d, d = 512$.

All the inputs from the datasets are normalized with the $\ell_2$ norm of 1. Note that We drop some date points compared with [11] to make the number of training data an integer multiple of the maximum batch size in our experiment, which is 512.

## N.3  Experiment Settings

**Hardware and Frameworks** All the experiments run with PyTorch=2.1.2 [36] and numpy=1.24.3 [37]. The codes run on a server with a single NVIDIA RTX 6000 GPU with AMD EPYC 7763 64-Core Processor.

**Problem Formulation** Given a binary classification task $\mathcal{D} = \{\mathbf{x}_i \in \mathbb{R}^d, y_i \in \{-1, +1\}\}_{i=1}^n$, our goal is to obtain a set of parameters $\mathbf{w}$ that optimizes the objective below:

$$\mathcal{L}(\mathbf{w}; \mathcal{D}) = \frac{1}{n}\sum_{i=1}^n l(\mathbf{w}^\top \mathbf{x}_i, y_i) + \frac{\lambda}{2}\|\mathbf{w}\|_2^2, \tag{88}$$

where the objective consists of a standard logistic regression loss $l(\mathbf{w}^\top x_i, y_i) = -\log \sigma(y_i \mathbf{w}^\top \mathbf{x}_i)$, where $\sigma(t) = \frac{1}{1+\exp(-t)}$ is the sigmoid function; and a $\ell_2$ regularization term where $\lambda$ is a hyperparameter to control the regularization, and we set $\lambda$ as $10^{-6} \times n$ across all the experiments. By simple algebra one can show that [7]

$$\nabla l(\mathbf{w}^\top \mathbf{x}_i, y_i) = (\sigma(y_i \mathbf{w}^\top \mathbf{x}_i) - 1) y_i \mathbf{x}_i + \lambda \mathbf{w}, \tag{89}$$

$$\nabla^2 l(\mathbf{w}^\top \mathbf{x}_i, y_i) = \sigma(y_i \mathbf{w}^\top \mathbf{x}_i)(1 - \sigma(y_i \mathbf{w}^\top \mathbf{x}_i)) \mathbf{x}_i \mathbf{x}_i^T + \lambda I_d. \tag{90}$$

Due to $\sigma(y_i \mathbf{w}^\top \mathbf{x}_i) \in [0, 1]$, it is not hard to see that we have smoothness $L = 1/4 + \lambda$ and strong convexity $\lambda$. The constant meta-data of the loss function in equation (88) above for the two datasets is shown in the table below:

Table 1: The constants for the loss function and other calculation on MNIST and CIFAR-10.

|  | expression | MNIST | CIFAR10 |
|---|---|---|---|
| smoothness constant $L$ | $\frac{1}{4} + \lambda$ | $\frac{1}{4} + \lambda$ | $\frac{1}{4} + \lambda$ |
| strongly convex constant $m$ | $\lambda$ | 0.0112 | 0.0097 |
| Lipschitz constant $M$ | gradient clip | 1 | 1 |
| RDP constant $\delta$ | $1/n$ | 8.8778e-5 | 0.0001 |

The per-sample gradient with clipping w.r.t. the weights $\mathbf{w}$ of the logistic regression loss function is given as:

$$\nabla_{clip} l(\mathbf{w}^\top \mathbf{x}_i, y_i) = \Pi_{\mathcal{C}_M} \left( (\sigma(y_i \mathbf{w}^\top \mathbf{x}_i) - 1) y_i \mathbf{x}_i \right) + \lambda \mathbf{w}, \tag{91}$$

where $\Pi_{\mathcal{C}_M}$ denotes the gradient clipping projection into the Euclidean ball with the radius of $M$, to satisfy the Lipschitz constant bound. According to Proposition 5.2 of [22], the per-sampling clipping operation still results in a $L$-smooth, $m$-strongly convex objective. The resulting stochastic gradient update on the full dataset is as follows:

$$\frac{1}{n} \sum_{i=1}^{n} \nabla_{clip} l(\mathbf{w}^T \mathbf{x}_i, y_i), \tag{92}$$

Finally, we remark that in our specific case since we have normalized the features of all data points (i.e., $\|x\| = 1$), by the explicit gradient formula we know that $\|(\sigma(y_i \mathbf{w}^\top \mathbf{x}_i) - 1) y_i \mathbf{x}_i\| \leq 1$.

**Learning from scratch set-up** For the baselines and our PNSGD unlearning framework, we all sample the initial weight $\mathbf{w}$ randomly sampled from i.i.d Gaussian distribution $\mathcal{N}(\mu_0, \frac{2\sigma^2}{m})$, where $\mu_0$ is a hyper-parameter denoting the initialization mean and we set as 0. For the PNSGD unlearning method, the burn-in steps $T$ w.r.t. different batch sizes are listed in Table. 2. we follow [11] and set $T = 10,000$ for the baselines (D2D and Langevin unlearning) to converge.

Table 2: The Burn-in step $T$ set for different batch sizes for the PNSGD unlearning framework

| batch size | 32 | 128 | 512 | full-batch |
|---|---|---|---|---|
| burn-in steps | 10 | 20 | 50 | 1000 |

**Unlearning request implementation.** In our experiment, for an unlearning request of removing data point $i$, we replace its feature with random features drawn from $\mathcal{N}(0, I_d)$ and its label with a random label drawn uniformly at random drawn from all possible classes. This is similar to the DP replacement definition defined in [38], where they replace a point with a special *null* point $\perp$.

**General implementation of baselines**

**D2D [9]:**

• Across all of our experiments involved with D2D, we follow the original paper to set the step size as $2/(L + m)$.

• For the experiments in Fig. 3a, we calculate the noise to add after gradient descent with the non-private bound as illustrated in Theorem. O.1 (Theorem 9 in [9]); For experiments with sequential unlearning requests in Fig. 3b, we calculate the least step number and corresponding noise with the bound in Theorem. O.2(Theorem 28 in [9]).

• The implementation of D2D follows the pseudo code shown in Algorithm 1,2 in [9] as follows:

---

**Algorithm 2** D2D: learning from scratch

---

1: **Input**: dataset $D$
2: **Initialize** $\mathbf{w}_0$
3: **for** $t = 1, 2, \ldots, 10000$ **do**
4:      $\mathbf{w}_t = \mathbf{w}_{t-1} - \frac{2}{L+m} \times \frac{1}{n} \sum_{i=1}^{n} (\nabla_{clip} l(\mathbf{w}_{t-1}^T \mathbf{x}_i, y_i))$
5: **end for**
6: **Output**: $\hat{\mathbf{w}} = \mathbf{w}_T$

---

---

**Algorithm 3** D2D: unlearning

---

1: **Input**: dataset $D_{i-1}$, update $u_i$; model $\mathbf{w}_i$
2: **Update dataset** $D_i = D_{i-1} \circ u_i$
3: **Initialize** $\mathbf{w}'_0 = \mathbf{w}_i$
4: **for** $t = 1, \ldots, I$ **do**
5:      $\mathbf{w}'_t = \mathbf{w}'_{t-1} - \frac{2}{L+m} \times \frac{1}{n} \sum_{i=1}^{n} \nabla_{clip} l((\mathbf{w}'_{t-1})^T \mathbf{x}_i, y_i))$
6: **end for**
7: **Calculate** $\gamma = \frac{L-m}{L+m}$
8: **Draw** $Z \sim \mathcal{N}(0, \sigma^2 I_d)$
9: **Output** $\hat{\mathbf{w}}_i = \mathbf{w}'_{T_i} + Z$

---

The settings and the calculation of $I, \sigma$ in Algorithm. 3 are discussed in the later part of this section and could be found in Section. O.

**Langevin unlearning [11]**

We follow exactly the experiment details described in [11].

**General Implementation of PNSGD Unlearning (ours)**

• We set the step size $\eta$ for the PNSGD unlearning framework across all the experiments as $1/L$.

• The pseudo-code for PNSGD unlearning framework is shown in Algorithm. 1.

### N.4 Implementation Details for Fig. 3a

In this experiment, we first train the methods on the original dataset $\mathcal{D}$ from scratch to obtain the initial weights $\mathbf{w}_0$. Then we randomly remove a single data point ($S = 1$) from the dataset to get the new dataset $\mathcal{D}'$, and unlearn the methods from the initial weights $\hat{\mathbf{w}}$ and test the accuracy on the testing set. We follow [11] and set the target $\hat{\epsilon}$ with 6 different values as $[0.05, 0.1, 0.5, 1, 2, 5]$. For each target $\hat{\epsilon}$:

• For D2D, we set two different unlearning gradient descent step budgets as $I = 1, 5$, and calculate the corresponding noise to be added to the weight after gradient descent on $\mathcal{D}$ according to Theorem. O.1.

• For the Langevin unlearning framework [11], we set the unlearning fine-tune step budget as $\hat{K} = 1$ only, and calculate the smallest $\sigma$ that could satisfy the fine-tune step budget and target $\hat{\epsilon}$ at the same time. The calculation follows the binary search algorithm described in the original paper.

• For the stochastic gradient descent langevin unlearning framework, we also set the unlearning fine-tune step budget as $\hat{K} = 1$, and calculate the smallest $\sigma$ that could satisfy the fine-tune step budget and target $\hat{\epsilon}$ at the same time. The calculation follows the binary search algorithm as follows:

---

**Algorithm 4** PNSGD Unlearning: binary search $\sigma$ that satisfy $\hat{K}$ and target $\hat{\epsilon}$ budget

---

1: **Input**:target $\hat{\epsilon}$, unlearn step budget $K$, lower bound $\sigma_{\text{low}}$, upper bound $\sigma_{\text{high}}$
2: **while** $\sigma_{\text{high}} - \sigma_{\text{low}} > 1e - 8$ **do**
3:     $\sigma_{\text{mid}} = (\sigma_{\text{low}} + \sigma_{\text{high}})/2$
4:     call Alg. 5 to find the least $K$ that satisfies $\hat{\epsilon}$ with $\sigma = \sigma_{\text{mid}}$
5:     **if** $K \leq \hat{K}$ **then**
6:         $\sigma_{\text{high}} = \sigma_{\text{mid}}$
7:     **else**
8:         $\sigma_{\text{low}} = \sigma_{\text{mid}}$
9:     **end if**
10: **end while**
11: **return** K

---

---

**Algorithm 5** PNSGD Unlearning [non-convergent]: find the least $K$ satisfying $\hat{\epsilon}$

---

1: **Input**:target $\hat{\epsilon}$, $\sigma$, burn-in $T$, projection radius $R$
2: **Initialize** $K = 1, \epsilon > \hat{\epsilon}$
3: **while** $\epsilon > \hat{\epsilon}$ **do**
4:     $c = 1 - \eta m$
5:     $\varepsilon_1(\alpha) = \frac{\alpha(2R)^2}{2\eta\sigma^2}c^{2Tn/b}$
6:     $Z = \frac{1}{1-c^{n/b}}\frac{2\eta M}{b}$
7:     $\varepsilon_2(\alpha) = \frac{\alpha Z^2}{2\eta\sigma^2}c^{2kn/b} + 2Rc^{Tn/b}$
8:     $\varepsilon(\alpha) = \frac{\alpha-1/2}{\alpha-1}(\varepsilon_1(2\alpha) + \varepsilon_2(2\alpha))$
9:     $\epsilon = \min_{\alpha>1}[\varepsilon(\alpha) + \log(1/\delta)/(\alpha - 1)]$
10:    $K = K + 1$
11: **end while**
12: **Return** $K$

---

We set the projection radius as $R = 100$, and the $\sigma$ found is listed in Table. 3.

Table 3: $\sigma$ of PNSGD unlearning.

|          |            | 0.05   | 0.1    | 0.5    | 1      | 2      | 5      |
|----------|------------|--------|--------|--------|--------|--------|--------|
| CIFAR-10 | b=128      | 0.2165 | 0.1084 | 0.0220 | 0.0112 | 0.0058 | 0.0025 |
|          | full batch | 1.2592 | 0.6308 | 0.1282 | 0.0653 | 0.0338 | 0.0148 |
| MNIST    | b=128      | 0.0790 | 0.0396 | 0.0080 | 0.0041 | 0.0021 | 0.0009 |
|          | full batch | 0.9438 | 0.4728 | 0.0960 | 0.0489 | 0.0253 | 0.0111 |

### N.5 Implementation Details for Fig. 3b

In this experiment, we fix the target $\hat{\epsilon} = 1$, we set the total number of data removal as 100. We show the accumulated unlearning steps w.r.t. the number of data removed. We first train the methods from scratch to get the initial weight $\mathbf{w}_0$, and sequentially remove data step by step until all the data points are removed. We count the accumulated unlearning steps $K$ needed in the process.

• For D2D, According to the original paper, only one data point could be removed at a time. We calculate the least required steps and the noise to be added according to Theorem. O.2.

• For Langevin unlearning, we fix the $\sigma = 0.03$, and we let the model unlearn $[5, 10, 20]$ per time. The least required unlearning steps are obtained following [11].

• For Stochastic gradient descent langevin unlearning, we replace a single point per request and unlearn for 100 requests. We obtain the least required unlearning step $K$ following Corollary 3.7. The pseudo-code is shown in Algorithm. 6.

---

**Algorithm 6** PNSGD Unlearning: find the least unlearn step $K$ in sequential settings
---
1: **Input**:target $\hat{\epsilon}$, $\sigma$, total removal $S$
2: $K_{\text{list}} = []$
3: **for** i in range($S$) **do**
4:      **Initialize** $K_{\text{list}}[i-1] = 1$, $\epsilon > \hat{\epsilon}$
5:      **while** $\epsilon > \hat{\epsilon}$ **do**
6:          $\epsilon = \min_{\alpha > 1}[\varepsilon(\alpha, \sigma, b, K_{\text{list}}[i-1]) + \frac{\log(1/\delta)}{\alpha - 1}]$
7:          $K_{\text{list}}[i-1] = K_{\text{list}}[i-1] + 1$
8:      **end while**
9: **end for**
10: **Return** $K_{\text{list}}$
---

---

**Algorithm 7** $\varepsilon(\alpha, \sigma, b, K)$
---
1: **Input**:target $\alpha$, $\sigma$, removal batch size $b$ per time, $i$-th removal in the sequence
2: $c = 1 - \eta m$
3: **Return** $\frac{\alpha Z_\mathcal{B}(b)^2}{2\eta\sigma^2} c^{2Kn/b}$
---

---

**Algorithm 8** $Z_\mathcal{B}(b)$
---
1: $c = 1 - \eta m$
2: **return** $\frac{1}{1 - c^{n/b}} \frac{2\eta M}{b}$
---

### N.6 Implementation Details for Fig. 3c and 3d

In this study, we fix $S = 100$ to remove, set different $\sigma = [0.05, 0.1, 0.2, 0.5, 1]$ and set batch size as $b = [32, 128, 512, \text{full batch}]$. We train the PNSGD unlearning framework from scratch to get the initial weight, then remove some data, unlearn the model, and report the accuracy. We calculate the least required unlearning steps $K$ by calling Algorithm. 6.

## O    Unlearning Guarantee of Delete-to-Descent [9]

The discussion is similar to those in [11], we include them for completeness.

**Theorem O.1** (Theorem 9 in [9], with internal non-private state). *Assume for all $\mathbf{d} \in \mathcal{X}$, $f(x; \mathbf{d})$ is $m$-strongly convex, $M$-Lipschitz and $L$-smooth in $x$. Define $\gamma = \frac{L-m}{L+m}$ and $\eta = \frac{2}{L+m}$. Let the learning iteration $T \geq I + \log(\frac{2Rmn}{2M})/\log(1/\gamma)$ for PGD (Algorithm 1 in [9]) and the unlearning algorithm (Algorithm 2 in [9], PGD fine-tuning on learned parameters **before** adding Gaussian noise)*

*run with $I$ iterations. Assume $\epsilon = O(\log(1/\delta))$, let the standard deviation of the output perturbation gaussian noise $\sigma$ to be*

$$\sigma = \frac{4\sqrt{2}M\gamma^I}{mn(1-\gamma^I)(\sqrt{\log(1/\delta)+\epsilon} - \sqrt{\log(1/\delta)})}. \tag{93}$$

*Then it achieves $(\epsilon, \delta)$-unlearning for add/remove dataset adjacency.*

**Theorem O.2** (Theorem 28 in [9], without internal non-private state). *Assume for all $\mathbf{d} \in \mathcal{X}$, $f(x; \mathbf{d})$ is $m$-strongly convex, $M$-Lipschitz and $L$-smooth in $x$. Define $\gamma = \frac{L-m}{L+m}$ and $\eta = \frac{2}{L+m}$. Let the learning iteration $T \geq I + \log(\frac{2Rmn}{2M})/\log(1/\gamma)$ for PGD (Algorithm 1 in [9]) and the unlearning algorithm (Algorithm 2 in [9], PGD fine-tuning on learned parameters **after** adding Gaussian noise) run with $I + \log(\log(4di/\delta))/\log(1/\gamma)$ iterations for the $i^{th}$ sequential unlearning request, where $I$ satisfies*

$$I \geq \frac{\log\left(\frac{\sqrt{2d}(1-\gamma)^{-1}}{\sqrt{2\log(2/\delta)+\epsilon}-\sqrt{2\log(2/\delta)}}\right)}{\log(1/\gamma)}. \tag{94}$$

*Assume $\epsilon = O(\log(1/\delta))$, let the standard deviation of the output perturbation gaussian noise $\sigma$ to be*

$$\sigma = \frac{8M\gamma^I}{mn(1-\gamma^I)(\sqrt{2\log(2/\delta)+3\epsilon} - \sqrt{2\log(2/\delta)+2\epsilon})}. \tag{95}$$

*Then it achieves $(\epsilon, \delta)$-unlearning for add/remove dataset adjacency.*

Note that the privacy guarantee of D2D [9] is with respect to add/remove dataset adjacency and ours is the replacement dataset adjacency. However, by a slight modification of the proof of Theorem O.1 and O.2, one can show that a similar (but slightly worse) bound of the theorem above also holds for D2D [9]. For simplicity and fair comparison, we directly use the bound in Theorem O.1 and O.2 in our experiment. Note that [38] also compares a special replacement DP with standard add/remove DP, where a data point can only be replaced with a *null* element in their definition. In contrast, our replacement data adjacency allows *arbitrary* replacement which intuitively provides a stronger privacy notion.

**The non-private internal state of D2D.** There are two different versions of the D2D algorithm depending on whether one allows the server (model holder) to save and leverage the model parameter *before* adding Gaussian noise. The main difference between Theorem O.1 and O.2 is whether their unlearning process starts with the "clean" model parameter (Theorem O.1) or the noisy model parameter (Theorem O.2). Clearly, allowing the server to keep and leverage the non-private internal state provides a weaker notion of privacy [9]. In contrast, our PNSGD unlearning approach by default only keeps the noisy parameter so that we do not save any non-private internal state. As a result, one should compare the PNSGD unlearning to D2D with Theorem O.2 for a fair comparison.

## P  Unlearning Guarantees of Langevin Unlearning [11]

In this section, we restate the main results of Langevin unlearning [11] and provide a detailed comparison and discussion for the case of strongly convex objective functions.

**Theorem P.1** (RU guarantee of PNGD unlearning, strong convexity). *Assume for all $\mathcal{D} \in \mathcal{X}^n$, $f_\mathcal{D}$ is $L$-smooth, $M$-Lipschitz, $m$-strongly convex and $\nu_\mathcal{D}$ satisfies $C_{LSI}$-LSI. Let the learning process follow the PNGD update and choose $\frac{\sigma^2}{m} < C_{LSI}$ and $\eta \leq \min(\frac{2}{m}(1 - \frac{\sigma^2}{mC_{LSI}}), \frac{1}{L})$. Given $\mathcal{M}$ is $(\alpha, \varepsilon_0)$-RDP and $y_0 = x_\infty = \mathcal{M}(\mathcal{D})$, for $\alpha > 1$, the output of the $K^{th}$ PNGD unlearning iteration achieves $(\alpha, \varepsilon)$-RU, where*

$$\varepsilon \leq \exp\left(-\frac{1}{\alpha}\frac{2\sigma^2\eta K}{C_{LSI}}\right)\varepsilon_0. \tag{96}$$

**Theorem P.2** (RDP guarantee of PNGD learning, strong convexity). *Assume $f(\cdot; \mathbf{d})$ be $L$-smooth, $M$-Lipschitz and $m$-strongly convex for all $\mathbf{d} \in \mathcal{X}$. Also, assume that the initialization of PNGD*

satisfies $\frac{2\sigma^2}{m}$-*LSI. Then by choosing $0 < \eta \le \frac{1}{L}$ with a constant, the PNGD learning process is* $(\alpha, \varepsilon_0^{(S)})$-*RDP of group size $S \ge 1$ at $T^{th}$ iteration with*

$$\varepsilon_0^{(S)} \le \frac{4\alpha S^2 M^2}{m\sigma^2 n^2}(1 - \exp(-m\eta T)). \tag{97}$$

*Furthermore, the law of the PNGD learning process is $\frac{2\sigma^2}{m}$-LSI for any time step.*

Combining these two results, the $(\alpha, \varepsilon)$-RU guarantee for Langevin unlearning is

$$\varepsilon \le \exp\left(-\frac{m\eta K}{\alpha}\right)\frac{4\alpha S^2 M^2}{m\sigma^2 n^2}. \tag{98}$$

On the other hand, combining our Theorem 3.2, 3.3 and using the worst case bound on $Z_\mathcal{B}$ along with some simplification, we have

$$\varepsilon \le c^{2Kn/b}\frac{\alpha}{2\eta\sigma^2}\left(\frac{2\eta M}{(1-c)b}\right)^2 = (1-\eta m)^{2Kn/b}\frac{\alpha}{2\eta\sigma^2}\left(\frac{2M}{mb}\right)^2 = (1-\eta m)^{2Kn/b}\frac{2\alpha M^2}{\eta m^2\sigma^2 b^2}. \tag{99}$$

To make an easier comparison, we further simplify the above bound with $b = n$ and $1 - x < \exp(-x)$ which holds for all $x > 0$:

$$\varepsilon \le (1-\eta m)^{2Kn/b}\frac{2\alpha M^2}{\eta m^2\sigma^2 b^2} < \exp(-\frac{2Kn\eta m}{b})\frac{4\alpha M^2}{m\sigma^2 b^2}\times\frac{1}{2m\eta} = \exp(-2K\eta m)\frac{4\alpha M^2}{m\sigma^2 n^2}\times\frac{1}{2m\eta}. \tag{100}$$

Now we compare the bound (98) obtained by Langevin dynamic in [11] and our bound (100) based on PABI analysis. First notice that the "initial distance" in (100) has an additional factor $1/(2m\eta)$. When we choose the largest possible step size $\eta = \frac{1}{L}$, then this factor is $\frac{L}{2m}$ which is most likely greater than 1 in a real-world scenario unless the objective function is very close to a quadratic function. As a result, when $K$ is sufficiently small (i.e., $K = 1$), possible that the bound (98) results in a better privacy guarantee when unlearning one data point. This is observed in our experiment, Figure 3a. Nevertheless, note that the decaying rate of (100) is *independent* of the Rényi divergence order $\alpha$, which actually leads to a much better rate in practice. More specifically, for the case $n = 10^4$ (which is roughly our experiment setting). To achieve $(1, 1/n)$-unlearning guarantee the corresponding $\alpha$ is roughly at scale $\alpha \approx 10$, since $\log(n)/(\alpha - 1)$ needs to be less than 1 in the $(\alpha, \varepsilon)$-RU to $(\epsilon, \delta)$-unlearning conversion (Proposition N.2). As a result, the decaying rate of (100) obtained by PABI analysis is superior by a factor of $2\alpha$, which implies a roughly 20x saving in complexity for $K$ large enough.

**Comparing sequential unlearning.** Notably, another important benefit of our bound obtained by PABI analysis is that it is significantly better in the scenario of sequential unlearning. Since we only need standard triangle inequality for the upper bound $Z_\mathcal{B}^{(s+1)} = \min(c^{K_s n/b}Z_\mathcal{B}^{(s)} + Z_\mathcal{B}, 2R)$ of the $(s+1)^{th}$ unlearning request, only the "initial distance" is affected and it grows at most linearly in $s$ (i.e., for the convex only case). In contrast, since Langevin dynamic analysis in [11] requires the use of weak triangle inequality of Rényi divergence, the $\alpha$ in their bound will roughly grow at scale $2^s$ which not only affects the initial distance but also the decaying rate. As pointed out in our experiment and by the author [11] themselves, their result does not support many sequential unlearning requests. This is yet another important benefit of our analysis based on PABI for unlearning.

