# OpenReview forum: "Certified Machine Unlearning via Noisy Stochastic Gradient Descent"
_NeurIPS.cc/2024/Conference — NeurIPS 2024 poster_

### Official Review · Reviewer_MVBT · 2024-07-12

**Soundness:** 3
**Presentation:** 3
**Contribution:** 2
**Rating:** 4
**Confidence:** 2

**Summary:**

The paper studies a practically important problem of machine unlearning and provides rigorous statistical guarantees for unlearning using the tools from the differential privacy. More specifically, it is shown that, for strongly convex losses, running Projected SGD on the new (adjacent) dataset guarantees that the removed data point will be forgotten, that is the output of "unlearning" algorithm will be close to the output of the Projected SGD if data point on a new dataset.

**Strengths:**

The algorithm is simple and easy to understand. The presentation is good with nice illustrations of the concepts. All necessary background is provided. The theoretical claims look sound to the best of my knowledge, apart from some technical remarks mentioned below.

**Weaknesses:**

(Major)

My main concern is very conceptual. Isn't it trivial that running SGD (unlearning part) on a strongly convex loss with updated dataset will eventually converge to the same unique optimum of the new loss as if we have started training from scratch. I want to highlight that in the strongly convex case, the optimum is unique. Moreover, the initial distance to the optimum is forgotten exponentially fast for constant step-size SGD. Therefore, I cannot understand why the derived result in this paper is not trivial. In my view, as expected, the authors face a difficulty in the convex case (getting only vacuous result Corollary 3.9).

(Minor)

1. Why the same batches are used in the unlearning process, shouldn't batches be from the updated dataset? This is perhaphs just a notational issue.
2. It is unclear to me what is the output of Algorithm 1.
3. It is unclear what it means to satisfy $(\alpha, \varepsilon)$-RU **with $c = 1-\eta m$** in Theorem 3.2. There is no parameter $c$ in definition 2.3.
4. The quantity $h_{\#}\mu$ in Lemma F.2. is not defined. This lemma is used in Proposition F.3., which is one of the key tools in analysis.
5. $\mathcal Z_{\mathcal B}$ appearing first in Theorem 3.2 is not defined in the main part. I had to go to Appendix to understand what it is.
6. Figure 3 (c) is not informative. All lines and confidence intervals overlap.
7. Why the method is called "contractive noisy iteration"? A more conventional name like Projected SGD would be more appropriate.
8. It would be helpful to explain the role of the projection in the algorithm. Is it possible to employ it without projection?
9. In the abstract, "convexity assumption" should be changed to "strong convexity assumption" because the guarantees in the convex case are vacuous in this paper.

**Questions:**

n/a

**Limitations:**

The results are limited to strongly convex objectives. To make the problem not trivial, it is crucial to consider convex or PL losses.

---

> ### Author Rebuttal · Authors · 2024-08-04
>
> We thank reviewer MVBT for their insightful comments and suggestions. We address the weaknesses and questions below.
>
> **W1: ``Is the unlearning privacy bound for strongly convex settings trivial? Are our results for convex settings vacuous?``**
>
> *Compare to retraining:* We agree with reviewer MVBT that under strong convexity, the optimum is unique and SGD can approach that optimum exponentially fast. However, as we have discussed theoretically in lines 234-243 and demonstrated in experiments, our unlearning strategy does offer computational benefits compared to retraining when the initialization is not very close to the optimal solution. In Figure 3(a), we have shown that even unlearning with ***one*** epoch is sufficient to achieve good accuracy (utility) and a strong privacy guarantee ($\epsilon \approx 1$). In contrast, for a mini-batch size $b=128$, we need $20$ epochs to ensure the result converges reasonably well.
>
> *Convex only:* There seems to be a misunderstanding regarding our convex result in Corollary 3.9. The bound is ***not*** vacuous when the training epoch $T$ is not too large so that $\frac{2\eta MT}{b} < 2R$. This condition can be met if the model learns reasonably well with moderate $T$ and the projection diameter $2R$ is not set to be extremely small. For example, with $2R = 10$, $b = 128$, $\eta = 1$, and $M = 1$, we can train for at most $639$ ***epochs*** while still holding $\frac{2\eta MT}{b} < 2R$. A smaller step size $\eta$ would allow for even more epochs. Under this scenario, our approach still provides computational benefits compared to retraining from scratch when the initialization is not close to ***every*** (each corresponding to an adjacent dataset) optimum solution (set). We will clarify this further in our revision.
>
> *Compare to noiseless fine-tuning:* Instead of retraining from scratch to ensure exact unlearning, another potential approach would be running ***noiseless*** SGD and then applying output perturbation (i.e., publishing weights with additive noise) to mitigate privacy leakage, as done in prior work [9]. Note that such methods also require strong convexity for their privacy guarantees. However, there are several downsides to this strategy compared to unlearning with PNSGD (i.e., adding noise at every iteration, our approach). These include an inferior privacy-utility-complexity trade-off, as demonstrated by our experiments, and the issue of non-private hidden states discussed in Appendix N.
>
> **Q1: ``Question about mini-batch``**
>
> Note that the mini-batches only contain indices, and the underlying dataset is different with adjacent datasets $\mathcal{D},\mathcal{D}^\prime$. Since we denote the replacement notion of dataset adjacency, the size of all considered datasets is always $n$ so that the mini-batch sequences can be the same. The main reason to keep it the same is for the analysis purpose, see also our response to DXXZ, Q1.
>
> **Q2: ``The output of Algorithm 1``**
>
> We apologize for the confusion. The output is the last iterates of learning and unlearning processes, which are $x^0_T, y^0_K$ respectively. We will make it clear in our revision.
>
> **Q3: ``Meaning of Theorem 3.2``**
>
> We apologize for the confusion. Theorem 3.2 indicates that the value $c$ is determined by the step size $\eta$ and the strong convexity parameter $\mu$, and it is used solely to determine the bound of $\varepsilon$. This has no bearing on the definition of $(\alpha,\varepsilon)$-RU. We will clarify this point in our revised manuscript to avoid any ambiguity.
>
> **Q4: $h_{\sharp} \mu$`` is undefined.``**
>
> We apologize for this oversight. For a function $h$ and a distribution $\mu$, $h_{\sharp} \mu$ denotes the pushforward operation. Essentially, it represents the distribution of $h(X)$, where the random variable/vector $X \sim \mu$. We will ensure this concept is properly defined and explained in the revision.
>
> **Q5: ``Question about ``$Z_{\mathcal{B}}$**
>
> We apologize for the confusion. The meaning of $Z_{\mathcal{B}}$ is explained in line 59, Figure 1, and its caption; it represents the upper bound of the $W_\infty$ distance between two adjacent learning processes. In Theorem 3.2, we provide an explicit formulation of $Z_{\mathcal{B}}$. We will ensure the meaning of $Z_{\mathcal{B}}$ is clear throughout our revised manuscript.
>
> **Q6: ``Why the name 'contractive noisy iteration'?``**
>
> We followed the terminology used in prior work [15, 16, 20], where the authors introduced the key technical tool (Lemma 2.6) that we employ in our analysis. To give due credit to their contribution, we decided to retain the same name.
>
> **Q7: ``The role of projection and is it necessary?``**
>
> We leverage the projection operation in multiple aspects of our analysis. Firstly, it is used to prove Theorem 3.1, which shows that the limiting distribution of the PNSGD learning process exists, is unique, and stationary. Secondly, we use it to bound the $W_\infty$ distance between the initial distribution and the target stationary distribution $\nu_{\mathcal{D}|\mathcal{B}}$. This bounding is crucial when applying our results without assuming that the learning process has attained its stationary distribution. This is precisely the aim of Theorem 3.2. If such an assumption is made, we can simplify the bound in Theorem 3.2, leading to Corollary 3.7. Note that the $Z_{\mathcal{B}}$ bound in Corollary 3.7 can always use the former term as an upper bound, rendering everything independent of $2R$, the projection diameter.
>
> In summary, if we directly assume the existence of the stationary distribution of the PNSGD learning process and that it is attained after learning, the projection operation can be omitted. However, in the general case, the projection is essential for the rigor and completeness of our analysis.

---

### Official Review · Reviewer_DXXZ · 2024-07-13

**Soundness:** 4
**Presentation:** 4
**Contribution:** 3
**Rating:** 7
**Confidence:** 3

**Summary:**

The paper proposes an effective and efficient machine unlearning algorithm based on projected noisy gradient descent (PSGD). The proposed methods can be extended to handle multiple unlearning requests.   The theoretical unlearning guarantee is established when the loss is assumed to be convex and smooth. Experiments verify the effectiveness of the proposed method.

**Strengths:**

1. A simple unlearning algorithm is established by PSGD, whose effectiveness and efficiency is verified by experiments. Especially, the experiments show that the proposed algorithm outperforms the baseline with less gradient complexity.

2. Theoretical analysis of unlearning guarantee for the proposed algorithm is established for convex smooth losses,  privacy-utility-complexity trade-off regarding the mini-batch size b for approximate unlearning is highlighted.

3, The paper is well-written and easy to follow.

**Weaknesses:**

1. The smoothness of the loss is required, which might limit the algorithm's applicability.

2. In experiments, a logistic loss with $\ell_2$ regularization is considered. It would be better to provides some results on convex loss since the corresponding theory is given.

**Questions:**

1. In algorithm, mini-batch sequence B is fixed. Is it reasonable in practice or analysis?

2. The paper states that "A smaller batch size b leads to a better privacy loss decaying rate". This seems counterintuitive. Could authors give some explanation?

---

> ### Author Rebuttal · Authors · 2024-08-04
>
> We thank reviewer DXXZ for their positive and thoughtful comments. We address the weaknesses and questions below.
>
> **W1:``Smoothness of the loss is required``**
>
> We agree with reviewer DXXZ that smoothness assumptions can restrict the applicability of our approach, for which will also list it as a limitation in Appendix B in our revision. We are currently working on relaxing these assumptions in the follow-up works.
>
> **Q1: ``Is mini-batch sequence B to be fixed in Algorithm 1 necessary?``**
>
> It is currently for analysis purposes, where similar limitations persist in the DP research of PNSGD [22]. We conjecture that one can randomly sample the mini-batch sequence at the beginning of each epoch. Our intuition is that instead of taking expectation with respect to $\mathcal{B}$ at the end of the (un)learning process, we can take the expectation for each epoch where the stationary property of the distribution remains. Nevertheless, the rigorous analysis in this direction is left as an interesting future work.
>
> **Q2: ``Question about a smaller batch size $b$ leading to a better privacy loss decaying rate.``**
>
> There appears to be a misunderstanding here. We are not saying that a smaller batch size $b$ will lead to better ***privacy loss***; instead, it will lead to a better ***privacy decaying rate***. To clarify, let's examine the bounds presented in Corollary 3.7, where the privacy loss is determined by the initial distance $Z_{\mathcal{B}}$ and the privacy decaying factor $c^{2Kn/b} = (c^{2n/b})^K$ for $K$ unlearning epochs. Notably, a smaller $b$ will lead to a better privacy decaying rate $c^{2n/b}$ ($c < 1$), since we are running more contractive updates (see Figure 1, green part) per epoch.
>
> However, as discussed in lines 234-243, an excessively small $b$ can increase the initial distance $Z_{\mathcal{B}}$. This is because $Z_{\mathcal{B}} = O(((1 - c^{n/b})b)^{-1})$, which is not monotonically decreasing with respect to $b$. Consequently, setting $b$ to its minimum value, such as $b = 1$, does not necessarily yield the optimal privacy guarantee. While smaller batch sizes improve the decaying rate, they can also adversely affect the initial privacy loss, necessitating a balanced choice of $b$, let alone its effect on the utility as discussed in line 230 and demonstrated in experiment section line 362.

---

### Official Review · Reviewer_SMx2 · 2024-07-15

**Soundness:** 3
**Presentation:** 3
**Contribution:** 3
**Rating:** 6
**Confidence:** 4

**Summary:**

The paper presents a simple scheme for machine unlearning. The algorithm has two parts: in the learning part, the models learns using the original dataset D, and in the second part it unlearns using a neighboring dataset D’. The algorithm uses 1) same batches at every epoch, and same batches for both learning and unlearning, 2) Gaussian noise for both learning and unlearning. Based on Renyi differential privacy and the relation between Renyi unlearning and (ε, δ)-unlearning, the paper shows that the privacy parameter ε drops exponentially with every epoch.

**Strengths:**

Novel set of results

**Weaknesses:**

My main concern is the lack of utility bounds. It’s not hard to design an algorithm which guarantees privacy: just don’t use the dataset at all. What is the utility your algorithm achieves after K iterations of unlearning?

**Questions:**

The assumptions in Theorem 3.2 are rather strong. You assume bounded domain + smoothness + Lipschitzness + strong convexity. Please justify these assumptions.

The notation is sometimes confusing. For example, there is notation \nu_{T | B} and there is notation \nu_{D | B}. Line 177: \nu_{D | B}^{0, ‘} is undefined. In Lemma F.2, notation # is undefined.

Minor issues:
-- Is line 173 the definition of \nu_{D | B}? Please make it clear. The wording sounds like this is already defined, and I spent quite some time trying to find the definition.

-- Algorithm 1: please list algorithm’s parameters, such as η and σ

-- Line 146: PABI was not defined

-- Theorem 3.2: Z_B: you don’t actually use B

-- Appendix D: Lipschitzness (no s after z)

-- Line 548: “Proof” -> ”Proof of Theorem 3.2”

-- I think the presentation could be simplified by focusing on some specific value of α

-- C_R: Line 97 doesn’t match Line 169

-- Theorem 3.2: relationship between D’ and D is not mentioned.

---

> ### Author Rebuttal · Authors · 2024-08-04
>
> We thank reviewer SMx2 for their careful reading, positive assessment, and thoughtful comments. We address the weaknesses and questions below.
>
> **W1: ``No utility bound.``**
>
> We thank reviewer SMx2 for the thoughtful comments. We agree that utility is an important aspect of the unlearning problem, but we do not think a utility bound is always necessary for a literature on machine unlearning. While a privacy bound is crucial to prevent the worst-case scenario, it is common practice in the machine learning literature to empirically measure utility in an average or data-dependent manner by conducting extensive experiments (as seen in the seminal DP-SGD paper [13]). This is also true in previous unlearning literature [7,11].
>
> However, we acknowledge that there are situations where a formal utility guarantee for the worst-case scenario is important. In such cases, the utility bound for noisy gradient descent is established in the differential privacy literature under strong convexity assumptions; see, for instance, Section 5 of [ref 1]. We will mention this point in our revision to provide clarity on when and how utility bounds can be applied.
>
> ### Reference
>
> [ref 1] Differential privacy dynamics of langevin diffusion and noisy gradient descent, Chourasia et al., NeurIPS 2021.
>
> **Q1: ``Justifying the assumption``**
>
> While it is true that our set of assumptions—covering strong convexity, Lipschitz continuity or bounded gradient norm, and smoothness—appears restrictive, these assumptions hold significant practical relevance. They cannot be directly applied to the modern neural networks but they still encompass critical learning problems, including logistic regression, as empirically demonstrated in our experiments.
>
> It is important to underscore that these assumptions are not unique to our work. They are foundational within the existing literature on machine unlearning [7-10] and are similarly necessary for studies related to the differential privacy (DP) guarantees and convergence of hidden-state PNSGD [15, 16, 22]. Specifically, the leading analytical approach for hidden-state PNSGD in privacy contexts—namely, the privacy amplification by iteration (or shifted divergence) analysis—relies on these assumptions.
> We acknowledge that relaxing these constraints represents a meaningful direction for future research. Indeed, prior works [15, 16, 22] identify this as an open problem, and we are actively conducting research to address these limitations.
>
> **Q2:``Clarity of notations and typos``**
>
> We thank reviewer SMx2 for their careful reading and suggestion. We will make sure all notations are defined and typos are corrected in our revision.
>
> **Q3:``Questions about Theorem 3.2``**
>
> Thanks for pointing out the issue regarding $\mathcal{Z}_{\mathcal{B}}$. Originally we derived a $\mathcal{B}$ dependent bound $\mathcal{Z}\_{\mathcal{B}}$ by Lemma 3.3. There we have $j_0$ that depends on when the differing index between $\mathcal{D},\mathcal{D}^\prime$ we encounter for a given mini-batch sequence $\mathcal{B}$. In the current Theorem 3.2, we directly choose the worst case $\mathcal{B}$, for which the bound is still valid (but loose). We will correct it in our revision and thank you again for spotting this issue.

---

### Decision · Program_Chairs · 2024-09-25

**Decision:**

Accept (poster)

**Comment:**

The paper considers the problem of machine unlearning and proposes to use a projected noisy stochastic gradient descent algorithm (PNSGD). This algorithm is shown to achieve unlearning for convex and strongly convex functions. While the results are mainly theoretical in nature, the authors complement their theoretical results with a small set of proof of concept experiments. The topic is interesting and relevant, however reviewers raise a few concerns. These concerns include the strong convexity assumption and authors rebut that many prior theoretical papers do make this assumption. Another big concern is the lack of utility bound. The authors argue that the utility bounds can largely be derived from the work of Chourasia et al. (2021), a point I concur with.

I strongly encourage authors to address reviewer comments in the final version of the paper. In particular, kindly add the utility bound and compare it with some of the prior works with the same set of assumptions e.g., Sekhari et al., 2021.